# Beneficial and detrimental consequences of AHR activation in intestinal infection

Oscar E Diaz[1], Liang Zhou[2], Christopher Barrington[1], Dennis Lindqvist[3], Frederike Graelmann[4], Emma Wincent[5], Brigitta Stockinger[1]

**The ligand-dependent transcription factor aryl hydrocarbon receptor (AHR) is an environmental sensor whose activation can have physiologically beneficial or detrimental consequences for host immune responses depending on the ligand. Here, we investigated the hypothesis that prolonged AHR activation either because of inefficient ligand metabolism or because of genetic manipulation may underlie the distinction between beneficial and detrimental effects. Our data indicate that prolonged AHR activation caused toxic endpoints for liver and thymus but was not per se interfering with the host response to infection with the intestinal pathogen *C. rodentium*. Genetically driven constitutive AHR activation improved resistance to infection, whereas prolonged AHR activation by the pollutant TCDD resulted in delayed clearance of *C. rodentium* associated with a suppression in antibody production. Combined single-cell RNA-seq and ATAC-seq analysis provided evidence that TCDD, but not genetic AHR activation, negatively affected dendritic cell functions such as activation, maturation, and antigen presentation. Thus, the detrimental impact of environmental pollutants such as TCDD on immune responses cannot solely be attributed to aberrantly prolonged activation of AHR.**

## Introduction

The aryl hydrocarbon receptor (AHR) is a ligand-dependent transcription factor that not only functions as a sensor of environmental pollutants but also recognizes dietary and microbial ligands (McMillan & Bradfield, 2007; Hubbard et al, 2015). It is evolutionarily conserved in most bilaterian animals with important roles in development of sensory neurons in invertebrates such as *Drosophila* and *Caenorhabditis elegans* (Hahn, 2002; Hahn et al, 2017). In vertebrates, research on AHR was initially focused on its role in the toxicity of man-made pollutants such as polychlorinated biphenyls (PCBs), dioxins, and polycyclic aromatic hydrocarbons (PAHs), and their detoxification by induction of biotransformation enzymes such as the cytochrome P450 family 1 (Cyp1). The toxicities were associated with exposure to dioxins such as 2,3,7,8-tetrachlorodibenzo-*p*-dioxin (TCDD) and related compounds (Poland & Knutson, 1982); however, they also pointed to important physiological functions in addition to inducing biotransformation enzymes.

It is now clear that the AHR has important functions, for instance, in mucosal sites such as the lung and gastrointestinal tract, which are continuously exposed to ligands derived from external sources such as pollutants, diet, or endogenous ligands generated from tryptophan metabolism by the constituents of the microbiota (Stockinger et al, 2021). Nevertheless, there is no clear consensus why and how environmental pollutants affect physiological AHR functions.

Several studies have addressed the detrimental effects of AHR activation by TCDD during influenza infection of the lung, such as suppression of innate and adaptive immune responses (Boule et al, 2018; Houser et al, 2024). In contrast, exposure to the endogenous tryptophan derivative 6-formylindolo[3,2-b]carbazole (FICZ), which has similar affinity for AHR, did not have such effects on the viral immune response (Wheeler et al, 2014). One of the prevailing theories explaining such differences is that the detrimental effects of TCDD are due to abnormally prolonged AHR stimulation (Mitchell & Elferink, 2009; Bock, 2019). Although the endogenous ligand FICZ is rapidly metabolized by AHR-induced Cyp1 enzymes (Bergander et al, 2004; Wincent et al, 2009) and therefore only activates AHR during a short time window, TCDD is not a good substrate for these enzymes and causes AHR activation persisting over weeks in mice (Birnbaum, 1986).

In this study, we investigated whether prolonged AHR activation is indeed the root cause for detrimental effects in host responses to infection. For this, we used two distinct models of prolonged AHR activation: a genetic model of constitutive AHR activation (Ye et al, 2017) and the AHR ligand TCDD. The experimental model we studied was infection with the pathogen *Citrobacter rodentium* as

[1]The Francis Crick Institute, London, UK  [2]Department of Infectious Diseases and Immunology, College of Veterinary Medicine, University of Florida, Gainesville, FL, USA  [3]Department of Environmental Science, Stockholm University, Stockholm, Sweden  [4]Immunology and Environment, Life and Medical Sciences (LIMES) Institute, University of Bonn, Bonn, Germany  [5]Institute of Environmental Medicine, Karolinska Institute, Stockholm, Sweden

Correspondence: Emma.wincent@ki.se; Brigitta.stockinger@crick.ac.uk

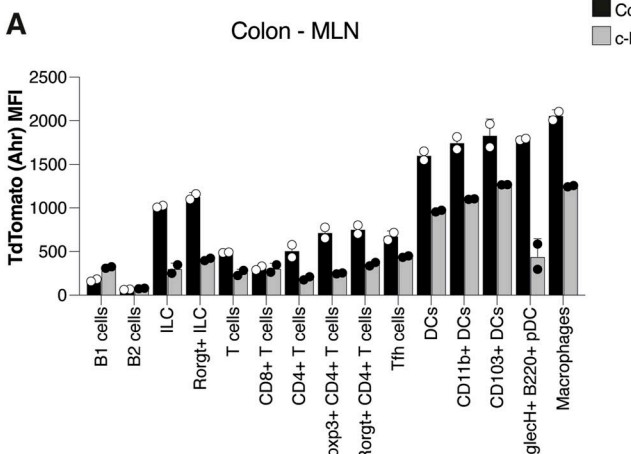

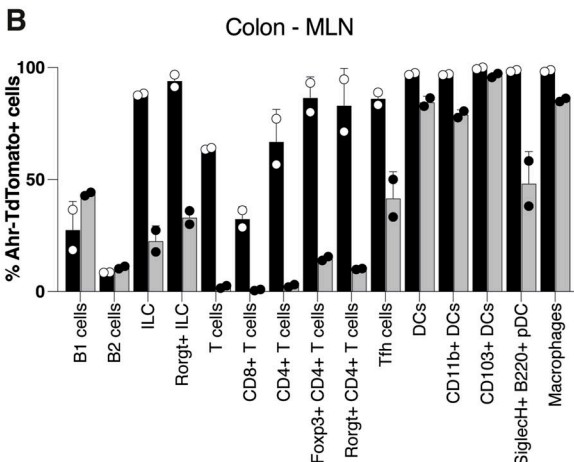

**Figure 1. AHR expression in intestinal immune cell populations.**
**(A)** AHR-TdTomato expression across immune cell types in the colon lamina propria and colon-draining mesenteric lymph node was determined by flow cytometry. Each dot represents the geometric MFI in the respective cell type. **(B)** Percentage of AHR-TdTomato–positive cells in each immune cell type shown in (A). Each dot represents the frequency of AHR-TdTomato+ cells in the respective cell type, n = 2 mice per group, and the bars show the mean + SD. Data are representative of three experiments.

our previous data had shown the importance of AHR in recovery from this infection (Schiering et al, 2017; Metidji et al, 2018).

Contrary to our expectations that prolonged AHR activation is detrimental to bacterial infections, mice with constitutive AHR activation were able to clear *C. rodentium* as efficiently as wild-type mice, whereas mice exposed to TCDD during infection showed delayed clearance and strongly reduced antibody responses to *C. rodentium*. Although previous studies showed that TCDD likely affects B cells directly (De Abrew et al, 2011), we suspected that TCDD might also affect a cell type that is required for induction of adaptive immune responses of both T and B cells, such as dendritic cells, as TCDD may disrupt dendritic cell homeostasis and function with consequences for the induction of T effector cells (Bankoti et al, 2010; Jin et al, 2014; Franchini et al, 2019). Using single-cell profiling of gene expression and chromatin accessibility of

immune cells from colon and colon-draining mesenteric lymph node (c-MLN), our data show that TCDD, but not constitutively active AHR, adversely affected the function of dendritic cells, in particular antigen presentation capacity and migration.

Thus, the duration of AHR signalling alone is not the decisive factor that characterizes the detrimental effect of AHR activation by TCDD. However, both scenarios of prolonged AHR activation produced the known toxic effects on the thymus, highlighting divergence of toxicity and effects on immune reactivity.

# Results

### Differential AHR expression on gut cell populations

AHR is widely expressed in the immune system in a cell type– and context-specific manner (Stockinger et al, 2014). To visualize AHR protein expression on a single-cell level by flow cytometry, we employed mice expressing a TdTomato fluorochrome knocked into the *Ahr* locus (Diny et al, 2022). Fig 1 shows a comparison of AHR levels and percentages on different immune cell types in colon and c-MLN (A, B). AHR expression levels were generally higher in cells from the colon, and the highest expression was seen in the myeloid lineage comprising dendritic cells and macrophages, confirming what was reported previously in the small intestine (Figs S1A and B and S2A and B) (Diny et al, 2022).

### Prolonged AHR activation in *Ahr*$^{dCAIR/dCAIR}$ and TCDD-treated mice

Next, we analysed the state of AHR activity in the genetic and TCDD model. In contrast to previous models of constitutively active AHR (CA-Ahr), which used a transgenic approach under the control of artificial genes and promoters outside of the *Ahr* locus (Andersson et al, 2002), the mouse model used here was generated using a knockin strategy into the endogenous locus. Specifically, a Flag-CA-Ahr-IRES-GFP (CAIR) allele preceded by a lox-stop-lox was introduced into the endogenous *Ahr* locus. Crossing with PGK-Cre led to the deletion of the STOP cassette so that mice expressed constitutively active AHR (dCAIR) in all tissues (Ye et al, 2017), hereafter referred to as *Ahr*$^{dCAIR/dCAIR}$. Homozygous *Ahr*$^{dCAIR/dCAIR}$ mice were used in all experiments and compared with wild-type (WT) B6 mice that received a single dose of 10 μg/kg TCDD by oral gavage.

Fig 2A shows fold change expression of the AHR target genes *Cyp1a1* and *Ahrr* in the colon of untreated *Ahr*$^{dCAIR/dCAIR}$ mice, indicative of constant AHR activation. Target gene expression in WT mice that had received TCDD 6 d before analysis was even higher (Fig 2B), and its expression along the gastrointestinal tract was constant, whereas vehicle-treated mice displayed a diminishing gradient of *Cyp1a1* expression from proximal small intestine to colon (Fig 2C).

We next compared the kinetics of *Cyp1a1* expression induced by TCDD in ileum, colon, and gonadal white adipose tissue (gWAT) over a period of 6 d with that after oral application of the proligand indole-3-carbinol (I3C), a dietary constituent that is converted to the high-affinity AHR ligand indolo[3,2-b]carbazole (ICZ) in the acidic environment of the stomach (Bjeldanes et al, 1991). The

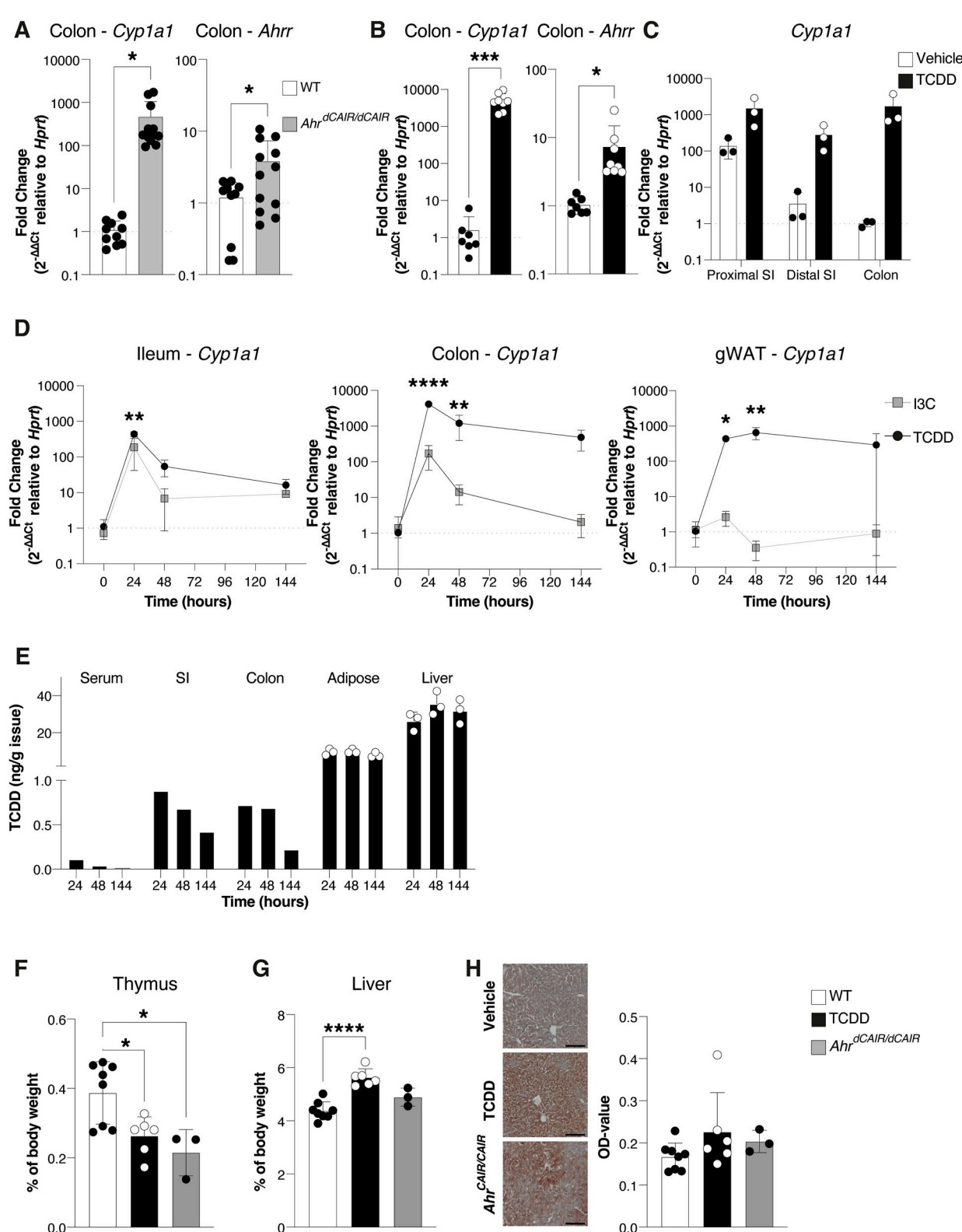

**Figure 2. Prolonged AHR activation.**
**(A, B, C)** *Cyp1a1* and *Ahrr* gene expression was determined by qRT-PCR from the indicated intestinal sections from *Ahr*^*dCAIR/dCAIR*^ mice and WT mice (A) and 6 d after a single administration of 10 μg/kg TCDD (B, C), and presented as fold change relative to *Hprt*. Each dot represents one mouse, and the bars show the mean + SD. **(A)** WT, n = 10 males; *Ahr*^*dCAIR/dCAIR*^ = 12 males; 3 experiments. **(B)** Vehicle, n = 7 males; TCDD = 8 males; 2 experiments. **(C)** Vehicle, n = 3 males; TCDD, n = 3 males; 1 experiment. Data in (C) are representative of two experiments. **(D)** *Cyp1a1* gene expression from the indicated organs was determined by qRT-PCR. Gonadal white adipose tissue = gWAT. Timepoints correspond to the hours after a single administration of 250 mg/kg I3C or 10 μg/kg TCDD in corn oil. Expression values were normalized to *Hprt* and to values from the vehicle-treated mice. Each dot represents the mean of 2–3 mice + SD. **(E)** TCDD levels detected in the indicated organ and timepoint after a single

rationale for including an assessment of *Cyp1a1* induction after I3C application was to have an internal comparison for AHR activation by the persistent ligand TCDD with a physiological ligand, which has similar affinity for AHR (Bjeldanes et al, 1991) but results in very transient activation because of its rapid clearance (Chen et al, 1995).

*Cyp1a1* levels induced by both TCDD and I3C application peaked at 24 h in the colon, but although oral I3C-induced *Cyp1a1* rapidly declined to baseline levels, TCDD-induced *Cyp1a1* expression remained constant, indicative of prolonged AHR signalling (Fig 2D). In adipose tissue, *Cyp1a1* induction was only seen after TCDD application and remained constant. The *Cyp1a1* induction strongly overlapped with accumulation of TCDD in respective tissue, which was most prominent in the adipose and liver as previously described (Gasiewicz et al, 1983), as compared to serum, SI, and colon (Fig 2E, Table S1). I3C or its AHR agonist derivate ICZ was not quantifiable in any of the analysed tissues and timepoints likely because of their short half-life (Chen et al, 1995; Anderton et al, 2004).

TCDD toxicity is associated with hepatomegaly, intrahepatic lipid accumulation, and thymic involution (Poland & Knutson, 1982). To assess whether prolonged AHR activation per se is responsible for toxicity characteristic of TCDD, we compared $Ahr^{dCAIR/dCAIR}$ mice with mice given a single dose of TCDD. Fig 2F–H shows that both models of prolonged AHR signalling exhibited thymic involution, whereas hepatomegaly and intrahepatic lipid accumulation were not evident in $Ahr^{dCAIR/dCAIR}$ mice. Lipid accumulation showed a trend for an increase in TCDD-treated mice but was not statistically significant.

### Effect of prolonged AHR activation on host response to infection

To evaluate the impact of prolonged AHR activation on the host response, we infected mice with the intestinal pathogen *C. rodentium*, which is a widely used model organism to study pathology and the host response to infection. There is a wealth of information about the kinetics of infection and the host innate and adaptive immune responses required to clear the pathogen (Basu et al, 2012; Caballero-Flores et al, 2021; Zindl et al, 2022). Furthermore, the importance of AHR signalling has been well documented in this model (Schiering et al, 2017; Metidji et al, 2018).

Upon infection with *C. rodentium*, $Ahr^{dCAIR/dCAIR}$ mice cleared the bacteria similar to wild-type mice (Figs 3A and S3A), made similar levels of IL-22 (Fig 3B), and exhibited lower inflammation indicated by faecal lipocalin-2 (Lcn2) levels (Fig 3C).

These data suggest that the prolonged activation of AHR in $Ahr^{dCAIR/dCAIR}$ mice did not negatively influence the host response against *C. rodentium* infection.

In contrast, mice treated with TCDD and infected with *C. rodentium* 6 d later exhibited delayed clearance of bacteria between day 7 and day 14, when most vehicle-treated mice had eliminated the pathogen (Fig 3D). These data were seen in male mice, but a similar response was observed in female mice (Fig S3B). We therefore used both sexes in subsequent experiments.

Despite the delayed bacterial clearance seen in TCDD-treated mice, they eventually also cleared the bacteria beyond day 21 (Fig S4). IL-22 levels were similar to controls in TCDD-treated mice (Fig 3E), and inflammation measured by lipocalin-2 faecal levels was comparable to that seen in infected mice that had not received TCDD (Fig 3F).

The data indicate that TCDD exposure, which similar to $Ahr^{dCAIR/dCAIR}$ causes prolonged AHR activation, had a detrimental effect on the host response to infection.

### Effect of prolonged AHR activation on humoral immunity

It is well established that a B-cell response of IgG antibodies to *C. rodentium* is essential for effective clearance of the pathogen (Maaser et al, 2004) and TCDD was previously reported to suppress B-cell responses (Vorderstrasse et al, 2001; Sherr & Monti, 2013; Boule et al, 2018; Houser & Lawrence, 2022). We therefore analysed the sera of mice exposed to TCDD 6 d before infection compared with infected vehicle-treated and $Ahr^{dCAIR/dCAIR}$ mice.

Sera analysed on day 14 after infection showed that TCDD-treated mice had lower antibody responses to *C. rodentium* than controls mainly affecting the IgM, IgG2b, and IgG2c subclasses, whereas IgG1, IgG3, and IgA responses were not affected (Figs 4A and S5A).

Conversely, antibody responses in $Ahr^{dCAIR/dCAIR}$ mice were comparable to control mice apart from the IgG2b response (Figs 4B and S5B). Although the antibody responses measured were quite variable, these results nevertheless suggest that TCDD has a broader suppressive effect than constitutive AHR activity in $Ahr^{dCAIR/dCAIR}$ mice. This was supported by similar reductions in T-dependent and T-independent antibody responses to unrelated antigens such as NP-CGG and NP-Ficoll after TCDD application (Fig S6A and B).

### Single-cell profiling highlights dendritic cell impairment by TCDD

Analysis of the c-MLN of TCDD-treated mice 14 d after infection indicated a decrease in total CD45⁺ immune cells, particularly evident for dendritic cell populations and B-cell subsets (Fig S7). These findings were reminiscent of previous reports in influenza infection of the lung after TCDD treatment (Vorderstrasse & Kerkvliet, 2001; Houser et al, 2024). Given the critical function of dendritic cells in inducing adaptive immune responses to infection, we therefore decided to embark on profiling single cells isolated from colon and c-MLN on day 5 after infection with *C. rodentium*, a time when the adaptive immune response is

---

administration of 10 μg/kg TCDD in corn oil normalized to the organ weight. Each dot represents one mouse for adipose and liver samples (n = 3 per group) and a pool of samples from three mice for the serum, small intestine (duodenum and ileum), and colon. **(F, G)** Weight of thymus and liver from $Ahr^{dCAIR/dCAIR}$ or TCDD-treated mice relative to body weight. Each dot represents one mouse, and the bars show the mean + SD. **(H)** Representative Oil Red O (ORO) staining in liver sections of $Ahr^{dCAIR/dCAIR}$ mice and TCDD-treated mice. Scale bars, 50 μm (left panel). Quantification of ORO stain (right panel). Each dot represents one mouse, and the bars show the mean + SD. **(F, G, H)** WT, n = 8 females; TCDD, n = 6 females; $Ahr^{dCAIR/dCAIR}$, n = 3 females; 2 experiments. *P < 0.05, **P < 0.01, ***P < 0.001, ****P < 0.0001. Unpaired *t* test (A, B), Two-way ANOVA with Šídák's multiple comparisons test. **(C, D)**, One-way ANOVA with Tukey's multiple comparisons test (F, G, H).

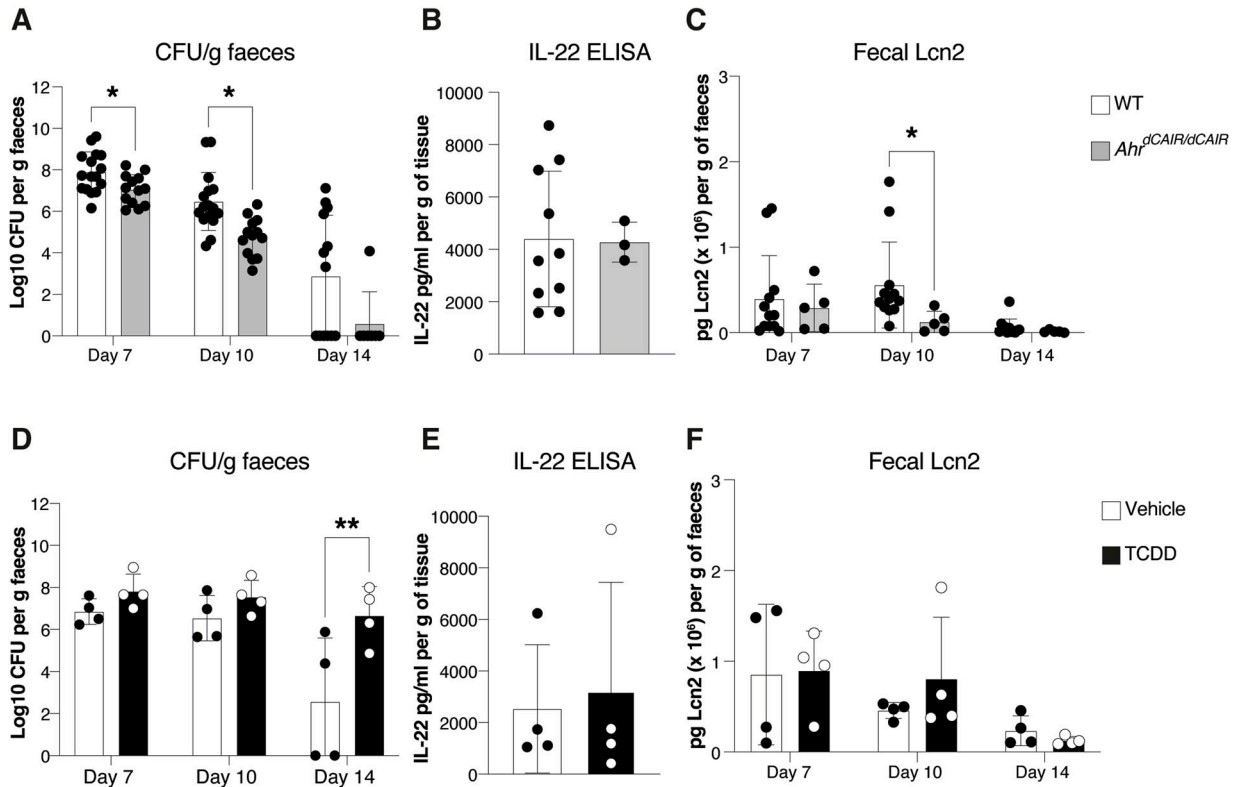

**Figure 3. Impact of prolonged AHR activation on *Citrobacter rodentium* infection.**
**(A, B, C)** *C. rodentium* burden in faecal pellets, (B) IL-22 protein levels in supernatants from colon explant cultures, (C) Lcn2 protein levels in faecal pellets from *Ahr^dCAIR/dCAIR* and WT mice. **(A, B, C)** WT, n = 16 males; *Ahr^dCAIR/dCAIR*, n = 13 males; 5 experiments; (B) WT, n = 10 males; *Ahr^dCAIR/dCAIR*, n = 3 males; 2 experiments; and (C) WT, n = 12 males; *Ahr^dCAIR/dCAIR*, n = 5 males; 3 experiments. **(D, E, F)** *C. rodentium* burden in faecal pellets, (E) IL-22 protein levels in supernatants from colon explant cultures and (F) Lcn2 protein levels in faecal pellets from TCDD-treated and vehicle-treated mice. Mice were infected 6 d after TCDD or vehicle administration. **(D, E, F)** Vehicle, n = 4 males; TCDD, n = 4 males; 1 experiment and representative of at least two experiments. Each dot represents samples collected from one mouse, and the bars show the mean + SD. Two-way ANOVA with Šidák's multiple comparisons test **(A, C, D, F)**, Unpaired *t* test (B, E). *$P < 0.05$, **$P < 0.01$.

activated. Our focus was on dendritic cells as the orchestrators for both arms of the adaptive immune response and in view of their exceptionally high AHR expression, which makes them likely targets for environmental factors acting through this receptor.

As dendritic cells are present in very low numbers in these tissues, we deliberately enriched the populations submitted to single-cell profiling for dendritic cells to increase their detection threshold. We performed simultaneous single-nucleus ATAC-seq and RNA-seq with a 1:1 mix of CD45+ cells and dendritic cells sorted from the colon and c-MLN of *Ahr^dCAIR/dCAIR* and TCDD-exposed mice, as well as vehicle-treated control mice at day 5 after *C. rodentium* infection (Fig 5A).

Clusters of the major immune cells and their different subpopulations were visualized using Uniform Manifold Approximation and Projection (UMAP) of the RNA-seq datasets. UMAPs generated for the RNA-seq and ATAC-seq datasets from colon and c-MLN of all conditions showed a similar representation of cell types (Fig S8A and B). Figs 5B and S8C give an overview of marker expression for identification of the major immune cell types found in vehicle-treated control mice in colon and c-MLN, respectively.

To identify the pathways modulated by *Ahr^dCAIR/dCAIR* mice and TCDD in cDC, we performed gene ontology analysis using differentially expressed genes (DEGs) in each condition compared with

the vehicle group. Figs 5C and S9 give an overview comparing TCDD and *Ahr^dCAIR/dCAIR* mice with vehicle in dendritic cells of colon and c-MLN. The most prominent difference was seen in pathways corresponding to antigen processing and presentation, which was markedly enhanced in dendritic cell subsets from *Ahr^dCAIR/dCAIR* mice, particularly in the colon compared with vehicle control, whereas genes in these pathways were down-regulated in dendritic cells from TCDD-treated mice in the c-MLN (Table S2). The populations of activated cDC1 and cDC2 characterized by high levels of *Ccr7*, *Cd40*, and *Relb* were not well separated in the colon and were therefore considered together.

There was no evidence of increased antigen processing and presentation pathways in c-MLN for *Ahr^dCAIR/dCAIR* when compared to vehicle, except in the subset of activated cDC1. Nevertheless, when compared to TCDD, both cDC1 and cDC2 in the c-MLN from *Ahr^dCAIR/dCAIR* mice showed an enrichment in up-regulated genes involved in antigen presentation (Fig S10, Table S3). Thus, the transcriptomic profiling confirmed the deficiency in antigen processing and presentation by dendritic cells from TCDD-treated mice and highlights enhancement of this function in *Ahr^dCAIR/dCAIR* mice above even vehicle-treated mice. This clearly indicates that prolonged AHR activation itself is not directly linked with suppression of the immune response.

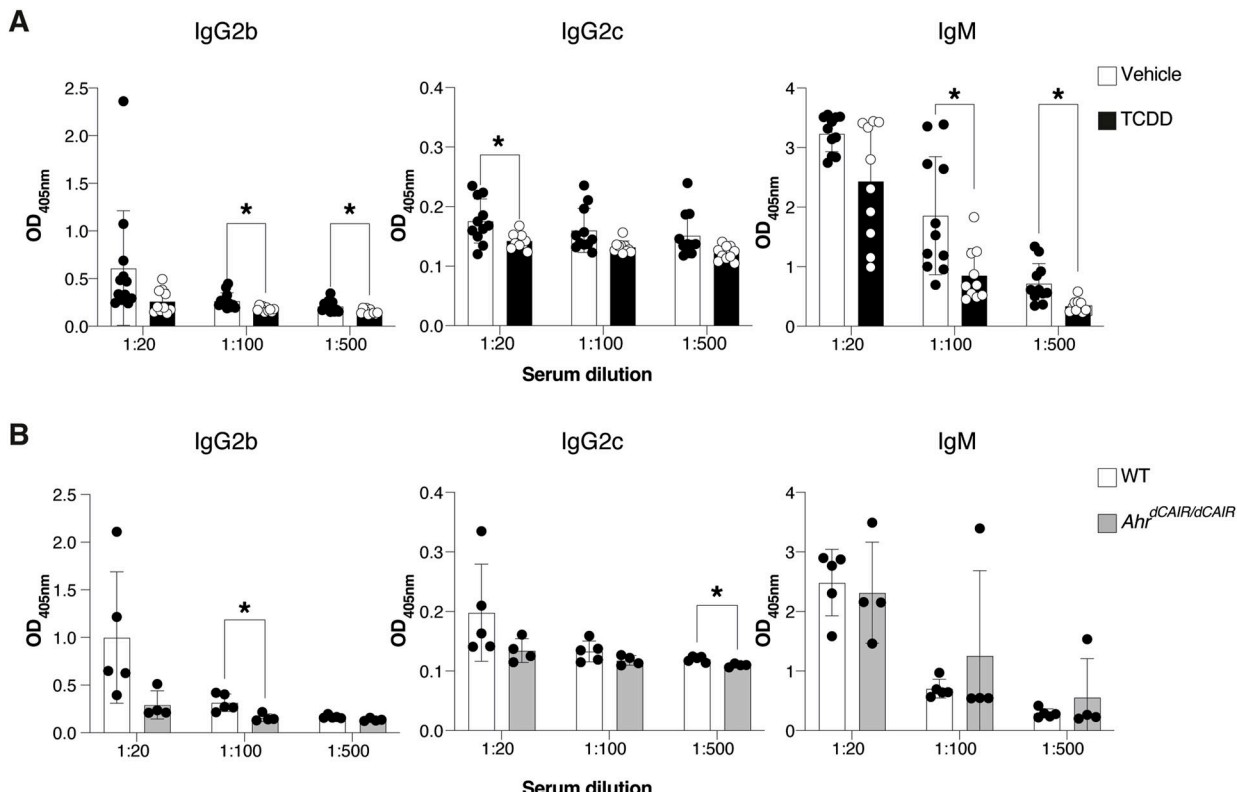

**Figure 4. TCDD reduces levels of *C. rodentium*-specific antibodies.**
**(A)** Levels of *C. rodentium*-specific Ig subclasses at the indicated serum dilutions 14 d after *C. rodentium* infection in mice that received 10 µg/kg TCDD or vehicle 6 d before infection. Vehicle, n = 6 females and 5 males; TCDD, n = 4 females and 6 males; 1 experiment. Data are representative of two experiments. Each dot represents samples collected from one mouse, and the bars show the mean + SD. **(B)** Levels of indicated Ig subclasses at the indicated serum dilutions 14 d after *C. rodentium* infection in *Ahr^{dCAIR/dCAIR}* and WT mice. WT, n = 3 females and 2 males; *Ahr^{dCAIR/dCAIR}*, n = 2 females and 2 males; 1 experiment. Data are representative of two experiments. Each dot represents one mouse, and the bars show the mean + SD. **(A, B)** Two-way ANOVA with Šídák's multiple comparisons test (A, B). *$P < 0.05$.

## Chromatin accessibility differences between TCDD-treated and *Ahr^{dCAIR/dCAIR}* mice

Ontology of differentially expressed genes and neighbouring regions with differentially accessible regions identified by ATAC-seq in dendritic cells from TCDD-treated compared with *Ahr^{dCAIR/dCAIR}* mice indicated small but discernible differences in genes linked to dendritic cell function. Examples of this are shown in Fig 6 where shaded areas highlight the peaks in question in colon cDC1 and activated cDC1 in the c-MLN, whereas violin plots show gene expression. Reduced accessibility and gene expression were seen in DC from TCDD-treated mice for amino acid transporter *Slc7a5* that plays an important role in the TORC1 pathway and influences DC/T-cell crosstalk, affecting maturation, migration, and antigen processing (Shao et al, 2024), and for *Hspa1b*, which encodes a heat shock protein that functions as chaperone with a role in the cellular stress response and protein quality control, including a role in antigen processing and presentation. Similarly, *H2-DMb1*, a gene located in the major histocompatibility class II locus that affects peptide binding to MHC class II in conjunction with chaperone cofactors, was less accessible in cDC1 from TCDD-treated mice. The final example is *Actb* encoding actin, vital for synapse formation between T cells and dendritic cells, as well as functions such as migration and antigen capture.

In activated cDC1 in c-MLN *Rhoc*, a member of the Rac subfamily of Rho GTPases, which are important for migration and adhesion, as well as the reorganization of the actin cytoskeleton (Ridley, 2006), was differentially accessible as was *Ifi30*, a lysosomal thiol reductase that plays a role in MHC class II–restricted antigen processing.

Overall, the ATAC-seq data confirm that TCDD but not constitutive AHR activation interfered with the induction of a group of genes that are required for crucial functions of dendritic cells such as migration, maturation, activation, and antigen presentation.

## Discussion

The increasing evidence for involvement of environmental factors in many inflammatory diseases has led to an interest in AHR as a potential therapeutic target. However, there is still a lack of clarity on what defines AHR agonists with beneficial properties compared with those that could cause toxicity and other adverse effects, which hampers progress with AHR-focused interventions.

The prevailing rationale for AHR-mediated toxicity seen at exposure to TCDD and other persistent xenobiotic AHR ligands has been associated with their prolonged AHR activation compared with rapidly metabolized endogenous ligands. The argument is

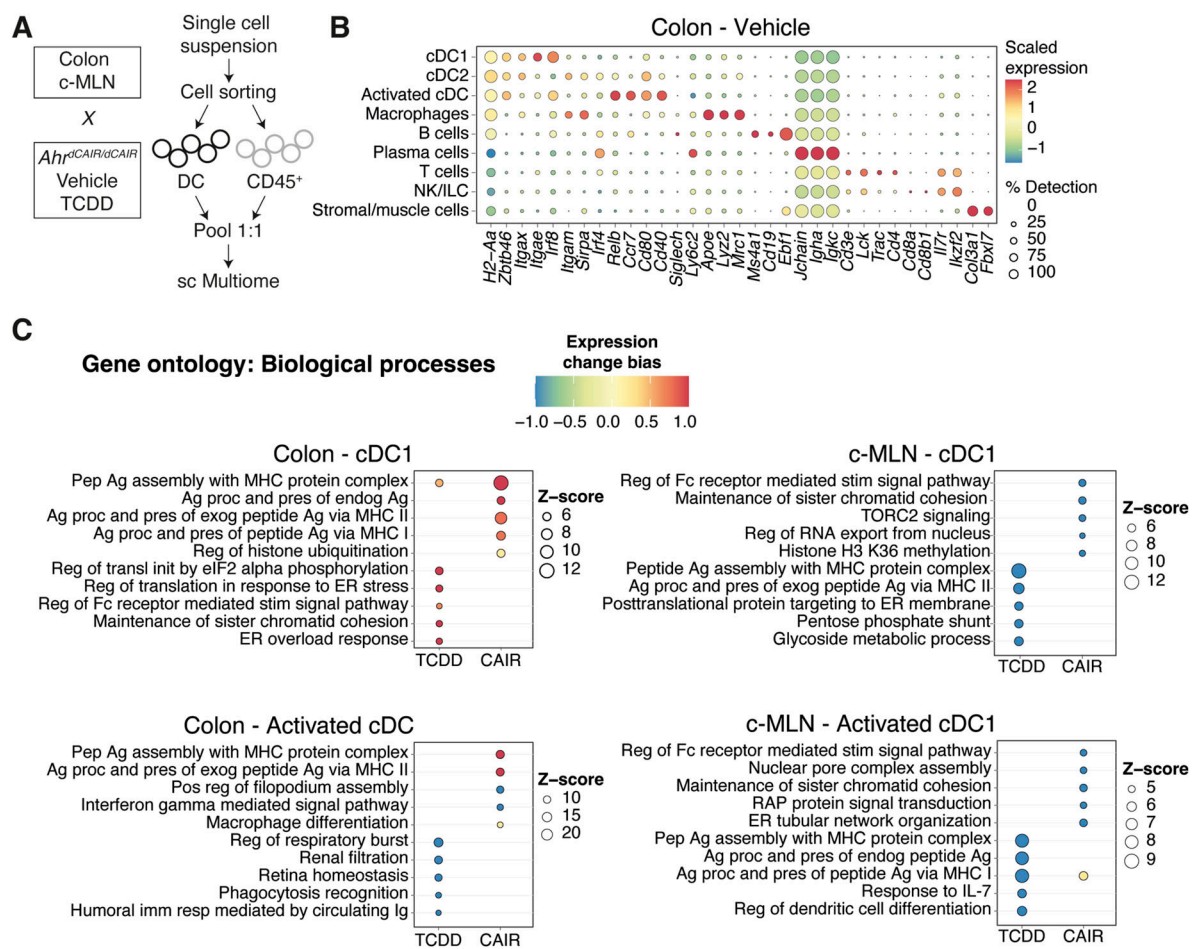

**Figure 5. Single-cell RNA-seq analysis.**
**(A)** Experimental setup for scMultiome from dendritic cells and CD45⁺ cells. **(B)** Dot plot showing the putative marker genes across cell types defined in the vehicle sample in the colon. The size of the dot represents the percentage of cells within clusters expressing the gene, whereas the colour scale represents the average expression level in that cluster. **(C)** Gene ontology enrichment analysis of biological processes for differentially expressed genes in TCDD and *Ahr^dCAIR/dCAIR* mice compared with vehicle (all males) for the indicated cell type. The colour scale represents the expression change bias, which represents the proportion of up-regulated and down-regulated genes that belong to that pathway, whereas the size of the dot shows the Z-score. Differentially expressed genes were defined as those with an adjusted *P*-value < 0.01, and enriched pathways have a *P*-value < 0.01. Abbreviations: Pep, peptide; Ag, antigen; proc, processing; pres, presentation; endog, endogenous; exog, exogenous; reg, regulation; pos, positive; transl, translation; init, initiation; stim, stimulatory; signal, signalling; imm, immune; resp, response; CAIR, *Ahr^dCAIR/dCAIR*.

persuasive as closer scrutiny of the AHR pathway highlights the emphasis on induction of negative feedback regulation after activation, such as the induction of Cyp1 family enzymes, as well as the AHR repressor (*Ahr*) (Chiaro et al, 2007; Vogel & Haarmann-Stemmann, 2017). Thus, one could assume that tight transient activation of the pathway is paramount for its physiological functioning. In this study, we tested whether prolonged activation of AHR per se is an inducement of detrimental activity that would perturb the normally beneficial AHR function in an intestinal infection. For this, we used TCDD and a knockin model (*Ahr^dCAIR/dCAIR*) to constitutively activate AHR under full control of the endogenous *Ahr* locus (Ye et al, 2017). *Ahr^dCAIR/dCAIR* mice showed a constitutive expression of *Cyp1a1* indicative of AHR activation, but no overt adverse effects were noticeable under steady-state conditions. This is in contrast to a previously published model of constitutive AHR activation in which a construct lacking the PAS-B domain was expressed as a transgene under a heterologous promoter,

resulting in the development of stomach cancer (Andersson et al, 2002). The *Ahr^dCAIR/dCAIR* mice did, however, replicate the thymus toxicitiy that has been reported after exposure to TCDD.

Given that the intestinal microenvironment is substantially influenced by AHR activity in multiple cell types, we chose the intestinal infection model with *C. rodentium* to probe the consequences of either TCDD application or endogenous constitutive AHR activity.

Against our initial assumption, *Ahr^dCAIR/dCAIR* mice showed no defects during this infection and dealt with it as efficiently as wild-type mice with less inflammation (Basu et al, 2012; Li et al, 2018; Melchior et al, 2024). In contrast, mice exposed to TCDD before infection clearly had a suboptimal response with delayed clearance and compromised antibody responses.

Thus, our data indicate that TCDD impairs the response of immune cells participating in the defence against *C. rodentium*. This has been observed in other organs, notably the lung, and the

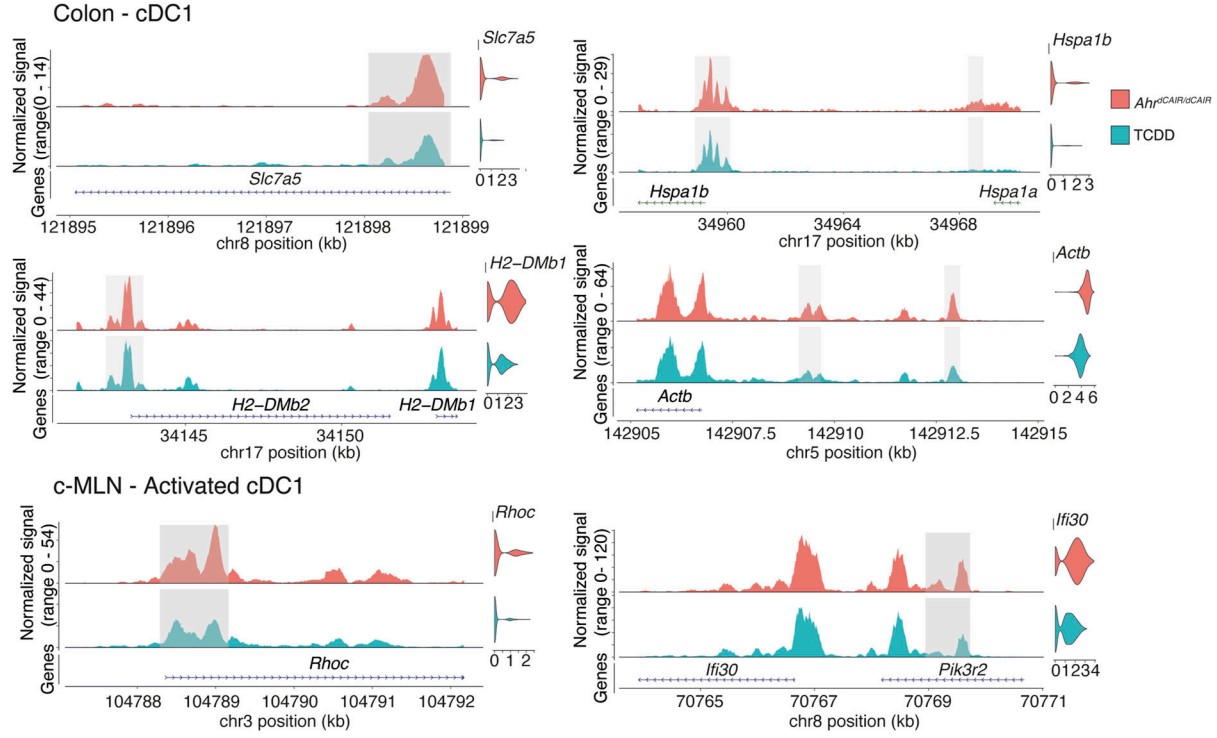

**Figure 6. Gene expression and chromatin accessibility in cDC1 of $Ahr^{dCAIR/dCAIR}$ mice compared with TCDD.**
ATAC-seq signal profiles of differentially accessible peaks and linked genes that are also differentially expressed in cDC1 populations in colon and MLN. Differentially accessible peaks are shaded in grey, whereas violin plots show gene expression in the respective population and condition. ATAC-seq profiles were generated from pseudo-bulk chromatin accessibility data, and violin plots show normalized expression values at the single-cell level.

toxicology literature refers to TCDD as an immunotoxic substance (Kerkvliet, 2002). Although the impairment of the B-cell response to *C. rodentium* by TCDD was not extensive, it did cause a delayed clearance of bacteria. The role of B cells in the host defence against *C. rodentium* is recognized but that of specific Ig isotypes is somewhat controversial (Maaser et al, 2004; Belzer et al, 2011). Here, the two Ig subclasses that were reduced after exposure to TCDD were IgG2b and IgG2c, which are important for complement fixation and opsonization of bacteria. IgG bound to *C. rodentium* virulence factors leads to their selective elimination by neutrophils (Kamada et al, 2015), and so it is conceivable that a reduction in this type of antibody may compromise speedy clearance of the pathogen. Furthermore, these two Ig subtypes are produced mainly by B1 cells and marginal zone B cells and are known to react against Gram-negative bacteria (Cerutti et al, 2013; Zeng et al, 2016). These cells also express the highest levels of AHR within the B-cell compartment (Fig 1 and [Villa et al, 2016]). It was notable that the numbers of dendritic cells and B cells were markedly decreased in TCDD-treated mice 14 d after infection, probably reflecting more a failure to expand rather than increased cell death. This is supported also by the observation that the lymph nodes of TCDD-treated mice were considerably smaller than those of vehicle-treated mice.

In order to get more mechanistic insight into the immune cell affected by TCDD and the comparison with constitutive AHR activation, we decided to carry out single-cell RNA sequencing combined with ATAC sequencing of cells isolated from colon and c-MLN on day 5 after infection. This is a timepoint when the

adaptive immune response to infection is beginning to take over from the initial innate response (Ahlfors et al, 2014; Zindl et al, 2022). This influenced the rationale to focus on dendritic cells for further in-depth analysis.

To our knowledge, this is also the first single-cell RNA-seq characterization of the immune compartment after *C. rodentium* infection, following a previous effort focused solely on CD4⁺ T cells (Kiner et al, 2021).

We enriched for dendritic cells to facilitate their profiling as they are normally present in very low numbers. Their obligatory role of T-cell activation supporting humoral and cellular immunity is well known, but it is becoming increasingly clear that B cells also may directly communicate with dendritic cells for effective antibody responses (Steiner et al, 2022). Our data highlight the markedly increased dendritic cell function in antigen processing and presentation seen in $Ahr^{dCAIR/dCAIR}$ mice in comparison with TCDD or vehicle-treated mice. When comparing dendritic cells from $Ahr^{dCAIR/dCAIR}$ mice to those from TCDD-treated mice, the enhancement in these functions became apparent in the c-MLN, indicating that the suppressive effect of TCDD on dendritic cell function is more evident in lymphoid organs where induction of immune responses takes place.

The overlay of ATAC-seq and RNA-seq also confirmed that chromatin in loci close to genes involved in migration, activation, and antigen processing/presentation was more accessible in dendritic cells from $Ahr^{dCAIR/dCAIR}$ than in those of TCDD-treated mice, coinciding with higher gene expression.

Taken together, our data confirm previous data in the literature, showing that TCDD results in suppression of immune responses. However, they also clearly indicate that prolonged AHR activation alone cannot account for this. Although the amplitude of AHR activation in *Ahr^{dCAIR/dCAIR}* mice was lower than that in TCDD-treated mice, the continuous state of activity nevertheless vastly exceeded what is seen in physiological AHR activation. Yet, mice with constitutively active AHR clearly had no impairment of their immune response to infection. One caveat, however, is the fact that *Ahr^{dCAIR/dCAIR}* mice exhibit prolonged AHR activity from birth and might have developed strategies to mitigate detrimental effects. It remains to be shown mechanistically what the underlying cause for the immunosuppressive effect of TCDD might be. Prolonged activation of AHR appears linked to known parameters of toxicity evident in liver and thymus, but in contrast may even enhance immune reactivity. It should be interesting to investigate how *Ahr^{dCAIR/dCAIR}* mice respond to different kinds of immune challenge.

# Materials and Methods

### Methods and protocols

A table listing reagents and tools is included.

**Reagents and tools table.**

| Reagent/Resource | Reference or source | Identifier or catalogue number |
|---|---|---|
| Experimental models | | |
| C57BL/6J mice | The Jackson Laboratory | Strain#:000664; RRID: IMSR_JAX:000664 |
| AHR-tdTomato mice | Francis Crick Institute | N/A |
| PGK-Cre mice | Francis Crick Institute | N/A |
| *Ahr^{CAIR}* mice | Ye et al (2017) | N/A |
| *Ahr^{dCAIR/dCAIR}* mice | This study | N/A |
| *C. rodentium* infection strain ICC180 | Dr.Gad Frankel, Imperial College London | N/A |
| Antibodies | | |
| CD16/CD32 monoclonal antibody (93), functional grade | eBioscience | Cat # 16-0161-86 |
| BD Horizon BUV496 rat anti-mouse CD45R/B220 (RA3-6B2) | BD Biosciences | Cat # 612950 |
| CD45R (B220) monoclonal antibody PE (RA3-6B2) | eBioscience | Cat # 12-0452-82 |
| Biotin anti-mouse CD3ε antibody (145-2C11) | BioLegend | Cat # 100304 |
| Brilliant Violet 711 anti-mouse CD3ε antibody (145-2C11) | BioLegend | Cat # 100349 |
| Biotin anti-mouse CD4 antibody (GK1.5) | BioLegend | Cat # 100403 |
| BD Horizon BV786 rat anti-mouse CD4 (RM4-5) | BD Biosciences | Cat # 563727 |
| Biotin anti-mouse CD8b.2 antibody (53–5.8) | BioLegend | Cat # 140406 |
| Biotin anti-mouse/human CD11b antibody | BioLegend | Cat # 101203 |
| PE/Cy7 anti-mouse/human CD11b antibody (M1/70) | BioLegend | Cat #101216 |
| APC anti-mouse CD11c antibody | BioLegend | Cat # 117310 |
| BV650 rat anti-mouse CD19 (1D3) | BD Biosciences | Cat # 563235 |
| PE/Dazzle 594 anti-mouse CD38 antibody (90) | BioLegend | Cat # 102729 |
| BD Horizon BUV395 rat anti-mouse CD45 (30-F11) | BD Biosciences | Cat # 565967 |
| Brilliant Violet 421 anti-mouse CD64 (FcγRI) antibody (X54-5/7.1) | BioLegend | Cat # 139309 |
| Brilliant Violet 605 anti-mouse CD90.2 (Thy1.2) antibody (30-H12) | BioLegend | Cat # 105343 |
| Brilliant Violet 510 anti-mouse CD103 antibody (2E7) | BioLegend | Cat # 121423 |
| BD Pharmingen biotin rat anti-mouse CD185 (CXCR5) (2G8) | BD Biosciences | Cat # 551960 |

**Continued**

| Reagent/Resource | Reference or source | Identifier or catalogue number |
|---|---|---|
| FOXP3 monoclonal antibody (FJK-16s), APC | eBioscience | Cat # 17-5773-82 |
| GL7 monoclonal antibody (GL-7 (GL7)), eFluor 450 | eBioscience | Cat # 48-5902-08 |
| Biotin anti-mouse Ly-6G/Ly-6C (Gr-1) antibody (RB6-8C5) | BioLegend | Cat # 108404 |
| BD Horizon BV786 rat anti-mouse IgD (11-26c.2a) | BD Biosciences | Cat # 563618 |
| MHC class II (I-A/I-E) monoclonal antibody, Alexa Fluor 700 (M5/114.15.2) | eBioscience | Cat # 56-5321-82 |
| MHC class II (I-A/I-E) monoclonal antibody, FITC, (M5/114.15.2) | eBioscience | Cat # 11-5321-82 |
| CD279 (PD-1) monoclonal antibody, Brilliant Ultra Violet 737 (J43) | eBioscience | Cat # 367-9985-82 |
| BD Horizon BV650 mouse anti-mouse RORγt (Q31-378) | BD Biosciences | Cat # 564722 |
| Siglec-H monoclonal antibody PerCP-eFluor 710 (eBio440c) | eBioscience | Cat # 46-0333-82 |
| Brilliant Violet 570 Streptavidin | BioLegend | Cat # 405227 |
| Biotin anti-mouse TER-119/erythroid cell antibody | BioLegend | Cat # 116204 |
| Oligonucleotides and other sequence-based reagents | | |
| *Hprt* | TaqMan assays | Cat # Mm00446968_m1 |
| *Cyp1a1* | TaqMan assays | Cat # Mm00487218_m1 |
| *Ahrr* | TaqMan assays | Cat # Mm00477443_m1 |
| Chemicals, enzymes, and other reagents | | |
| CountBright Absolute Counting Beads, for flow cytometry | Invitrogen | Cat # C36950 |
| DRAQ7 Dye | Invitrogen | Cat # D15105 |
| CD45 MicroBeads, mouse | Miltenyi Biotec | RRID:AB_2877061 |
| MACS BSA Stock Solution | Miltenyi Biotec | Cat # 130-091-376 |
| Digitonin | Thermo Fisher Scientific | Cat # BN2006 |
| Trizma hydrochloride solution, 100 ml | Sigma-Aldrich | Cat # T2194-100ML |
| NaCl (5 M), RNase-free | Invitrogen | Cat # AM9759 |
| Magnesium chloride solution | Sigma-Aldrich | Cat # M1028-100ML |
| IGEPAL CA-630, 50 ml | Sigma-Aldrich | Cat # I8896-50ML |
| DL-dithiothreitol solution | Sigma-Aldrich | Cat # 646563-10X.5ML |
| Protector RNase inhibitor | Roche | Cat # 3335399001 |
| 10% Tween-20 | Bio-Rad | Cat # 1662404 |
| Corning PBS, 1X without calcium and magnesium, PH 7.4 ± 0.1 | Corning | Cat # 21-040-CV |
| Collagenase from *Clostridium histolyticum* | Sigma-Aldrich | Cat # C2139-5G |
| DNase I | Roche | Cat # 10104159001 |
| UltraPure 0.5 M EDTA, pH 8.0 | Invitrogen | Cat # 15575020 |
| Hepes (1 M) | Gibco | Cat # 15630056 |
| Percoll | Cytiva | Cat # 17-0891-01 |
| HBSS (10X), no calcium, no magnesium, no phenol red | Gibco | Cat # 14185052 |
| HBSS (10X), calcium, magnesium, phenol red | Gibco | Cat # 14060040 |
| 2,3,7,8-Tetrachlorodibenzo-p-dioxin (TCDD) | AccuStandard | Cat # D-404N |
| Indole-3-carbinol (I3C) | Sigma-Aldrich | Cat # I7256-5G |
| Kanamycin bacterial plates | The Francis Crick Institute | N/A |

**Continued**

| Reagent/Resource | Reference or source | Identifier or catalogue number |
|---|---|---|
| LB medium | The Francis Crick Institute | N/A |
| Brilliance *E. coli*/Coliform Selective Medium (Dehydrated) | Thermo Fisher Scientific | Cat # CM1046B |
| Kanamycin sulphate | Sigma-Aldrich | Cat # K4000-25G |
| Dimethyl sulphoxide | Sigma-Aldrich | Cat # 41648-250ML |
| Corn oil | Sigma-Aldrich | Cat # C8267-500ML |
| Glacial acetic acid | Fisher Chemical | Cat # A/0360/PB17 |
| TRIzol reagent | Invitrogen | Cat # 15596018 |
| TaqMan Universal PCR Master Mix | Applied Biosystems | Cat # 4318157 |
| Micro sample tube Serum Gel CAT, 1.1 ml | Sarstedt | Cat # 41.1378.005 |
| NP-CGG (Chicken Gamma Globulin), Ratio 20–29 | LGC Genomics Ltd | Cat # N-5055C-1 |
| NP-AECM-FICOLL | LGC Genomics Ltd | Cat # F-1420-10 |
| Alhydrogel adjuvant 2% | InvivoGen | Cat # vac-alu-50 |
| Alkaline Phosphatase Yellow (pNPP) liquid substrate system for ELISA | Sigma-Aldrich | Cat # P7998-100ML |
| cOmplete, EDTA-free protease inhibitor cocktail | Roche | Cat # 11873580001 |
| GolgiStop | BD Biosciences | Cat # 554724 |
| FBS | Sigma-Aldrich | Cat # F7524-500ML |
| Sucrose | Sigma-Aldrich | Cat S5016-500G |
| Tissue-Tek O.C.T. compound | Sakura | Cat # 4583 |
| Mayer's haematoxylin solution | Sigma-Aldrich | Cat # MHS16-500ML |
| 10% neutral buffered formalin | Cell Path | Cat # BAF-0010-10P |
| VectaMount AQ Aqueous Mounting Medium (H-5501-60) | Vector Laboratories | Cat # H-5501-60 |
| Software | | |
| FlowJo | BD | https://www.flowjo.com/ |
| BD FACSDiva | BD | https://www.bdbiosciences.com/en-gb/products/software/instrument-software/bd-facsdiva-software |
| GraphPad Prism 10 | GraphPad | https://www.graphpad.com/ |
| EndNote | EndNote | https://www.endnote.com/ |
| R 4.4.1 | R Project for Statistical Computing | https://cran.r-project.org/ |
| R packages Seurat (5.0.1), Signac (1.14.0), intrinsicDimension (1.2.0), clustree (0.5.1), msigdbr (7.5.1) | CRAN | https://cran.r-project.org/ |
| R packages scDblFinder (1.18.0), clusterProfiler (4.12.6) | Bioconductor | https://www.bioconductor.org/ |
| Macs2 | Pypi | https://pypi.org/project/MACS2/2.2.7.1/ |
| Loupe Browser 8 | 10x Genomics | https://www.10xgenomics.com/support/software/loupe-browser/ |
| Illustrator | Adobe | https://www.adobe.com/uk/products/illustrator.html |
| Other | | |
| High-Capacity cDNA Reverse Transcription Kit | Applied Biosystems | Cat # 4368814 |
| Mouse IL-22 Uncoated ELISA Kit | Invitrogen | Cat # 88-7422-88 |
| Mouse lipocalin-2/NGAL DuoSet ELISA | R&D Systems | Cat # DY1857-05 |
| SBA Clonotyping System-C57BL/6-AP | SouthernBiotech | Cat # 5300-04B |

**Continued**

| Reagent/Resource | Reference or source | Identifier or catalogue number |
|---|---|---|
| LIVE/DEAD Fixable Near-IR Dead Cell Stain Kit | Invitrogen | Cat # L10119 |
| Foxp3/Transcription Factor Staining Buffer Set | eBioscience | Cat # 00-5523-00 |
| Transcription Factor Phospho Buffer Set (RUO) | BD Biosciences | Cat # 15844409 |
| BD FACSymphony A5 Cell Analyzer | BD Biosciences | N/A |
| Chromium Next GEM Single Cell Multiome ATAC + Gene Expression Reagent Kit | 10x Genomics | Cat # PN-1000283 |
| Luna-FX7 Automated Cell Counter | Logos Biosystems | Cat # L70001 |
| Acridine orange/propidium iodide stain | Logos Biosystems | Cat # L70001 |
| NovaSeq 6000 and NovaSeq X | Illumina | N/A |

## Mice

AHR-TdTomato (Diny et al, 2022), PGK-Cre (MGI ID:2178050), $Ahr^{CAIR}$ (Ye et al, 2017), $Ahr^{dCAIR/dCAIR}$ (all on a C57BL/6J background), and C57BL/6J mice used in this study were bred and maintained in individually ventilated cages under specific pathogen-free conditions at the Francis Crick Institute, according to protocols approved by the UK Home Office and the Ethics committee of the Francis Crick Institute. Mouse experiments were conducted according to the guidelines detailed in the Project Licenses granted by the UK Home Office to Brigitta Stockinger (PP0858308). Mice were age- and sex-matched and between 6- and 9-wk-old when first used. Both female and male mice were used. Exclusion criteria such as inadequate staining or low cell yield owing to technical problems were predetermined. Mice were randomly assigned to experimental groups.

## Infection with *C. rodentium*

For *C. rodentium* infection, a frozen stock of *C. rodentium* strain ICC180 (kindly provided by Dr. Gad Frankel, Imperial College London) was streaked overnight on a Luria-Bertani (LB) agar plate supplemented with 50 μg/ml kanamycin and incubated overnight in a dry incubator at 37°C. A single colony was picked and grown in LB broth supplemented with 50 μg/ml kanamycin and grown to log phase, followed by centrifugation and resuspension in PBS. Mice were orally gavaged with 200 μl of PBS containing $2 \times 10^9$ *C. rodentium* CFU. To determine the bacterial load, tissue pieces or faecal pellets were weighed and homogenized in sterile PBS and serial dilutions were plated onto LB plates supplemented with 50 μg/ml kanamycin or plates with Brilliance *E. coli*/Coliform Selective Medium and incubated overnight at 37°C. The numbers of CFU were normalized to the weight of the faecal pellets.

Sample sizes for experimental groups were determined based on available litter sizes or previous experience in the laboratory with the *C. rodentium* model. Animals with pathogen levels below $10^7$ CFU per gram of tissue on day 7 post-infection were excluded from the experimental results.

## Ligand administration

TCDD (AccuStandard) was dissolved at 100 μg/ml in DMSO by sonication for 30 min and stored at –80°C. For in vivo administration, TCDD was dissolved in corn oil at a concentration of 2 μg/ml and administered by oral gavage as a single dose of 10 μg/kg of body weight. To test the effect of TCDD on the clearance of *C. rodentium*, a single dose of 10 μg TCDD per kg was given 6 d before the infection. Vehicle-treated mice were given an equivalent dose of corn oil with 2% DMSO at a volume of 5 μl/g of body weight.

For in vivo administration, I3C (Sigma-Aldrich) was dissolved at a concentration of 50 mg/ml in 10% DMSO, 0.5% glacial acetic acid, and 89.5% corn oil. I3C was administered by oral gavage as a single dose of 250 mg/kg of body weight.

## Chemical tissue extraction

Extraction of TCDD was performed using a sequence of extraction and precipitation steps depending on the type of tissue. Preweighed intestinal tissues were homogenized frozen using pre-chilled 5-mm stainless steel beads and pre-chilled TissueLyser II (QIAGEN) at 30 Hz, in 30-s intervals. Liver and adipose tissues were homogenized in a similar way, but not frozen. Isopropanol was thereafter added to the homogenates and again run using the TissueLyser II for 30 s. Internal standard (PCB118) was added to the respective sample, and the tissue homogenates were extracted by vortexing, followed by centrifugation at 2,400g for 1 min. The isopropanol phase was transferred to a clean tube, and the pellet was re-extracted with 3-methylpentane (3MTP) by vortexing, followed by centrifugation at 2,400g, and transfer of the 3MTP supernatant to the isopropanol phase. Water containing 0.1 M sulphuric acid was added to the combined supernatants at an equal volume as the isopropanol, and the suspension was centrifuged at 2,400g for 1–2 min. The organic phase was thereafter transferred to a clean tube, and concentrated sulphuric acid was carefully added in a ventilated hood. The tubes were inverted until mixed, and centrifuged at 6,200g for 3–5 min, and the upper phase was finally transferred to an HPLC vial. Because of their low concentrations of TCDD, the replicate samples of intestinal tissue extracts were pooled and run through a sulphuric acid column as an additional

clean-up step before being analysed as described below. Serum samples were extracted the same way, starting from the step of adding isopropanol.

Extraction of I3C and ICZ was largely performed according to a previous study (Anderton et al, 2004). Briefly, frozen tissues were homogenized according to the method for TCDD, after which the pulverized tissues were mixed with PBS and re-homogenized using the TissueLyser II. Acetonitrile containing the internal standard (6-formylindolo[3,2-b]carbazole, FICZ) was added, and the samples were extracted by vortexing for 2–3 min. Samples were centrifuged at 2,400$g$ for 10 min, and the supernatants were transferred to clean tubes. The supernatants were evaporated to dryness in the presence of 15 $\mu$l DMSO under a stream of nitrogen gas, then resuspended in 10% acetonitrile, and centrifuged at 6,200$g$ for 5 min. Supernatants were transferred to HPLC vials before analysis as described below.

### Chemical analysis of TCDD, I3C, and ICZ

Analysis of TCDD was conducted on an Agilent 7890A GC (Agilent Technologies) with a multimode injector (MMI), coupled to a 5975C mass spectrometer (MS). The MS was operated in electron capture negative ionization (ECNI) mode. 1 $\mu$l was injected in splitless mode at 260°C. The split was opened after 2 min, and the injector temp was increased to 325°C, with a purge flow of 60 ml/min. The septum purge was 3 ml/min. Helium was used as a mobile gas at a constant flow of 1.5 ml/min. The GC oven was programmed from 80°C (held for 2 min) to 220°C at 40°C/min (held for 1 min), to 310°C at 40°C/min (held for 2.5 min). Separation was achieved on a J&W DB-5MS-UI capillary column (30 m × 0.25 mm i. d. × 0.25 $\mu$m film thickness; J&W Scientific). The transfer line to the MS was set at 310°C, the ion source temperature was 200°C, and the quadrupole temperature was 150°C. Methane was used as a buffer gas for ECNI/MS, and detection was done in selective ion mode (SIM). The internal standard PCB118 was detected using the ion m/z: 327.9, whereas TCDD was detected using the chlorine ions m/z: 35 and 37. Quantifications were conducted against external calibration curves.

Analysis of I3C and ICZ was conducted using a Biocompatible Agilent HPLC system, equipped with a 1290 Infinity II Bio Multi-sampler, 1260 Infinity II Bio Flexible Pump, Fluorescence Detector Spectra, and Diode Array Detector HS. The samples (50 $\mu$l) were injected onto the HPLC system, and separation was achieved using a Poroshell 120 EC-C18 column (3.0 × 50 mm, 2.7-micron particle size; Agilent InfinityLab). The mobile phase consisted of water (A) and acetonitrile (B), both with 1.5 mM formic acid. The gradient was as follows: 10% B to 50% B from 0 to 2 min, and 50% B to 70% B from 2 to 6 min, followed by re-equilibration to 10% for 4 min. The flow rate was 0.63 ml/min, and separation was carried out at 35°C. I3C and ICZ, as well as the internal standard FICZ, were monitored by both UV detection (280/254 nm) and fluorescence detection (390: 518-nm excitation/emission), and quantified against external calibration curves.

### Liver and thymus toxicity analysis

6- to 7-wk-old mice were given corn oil or a single dose of 10 $\mu$g/kg of body weight and analysed 6 d later. Mice were weighed before organ collection. Thymi were cleaned and weighed. Whole livers were collected and weighed, and a half of the left lobe was fixed in 10% neutral buffered formalin overnight. After washing twice in PBS, the organs were cryoprotected in 30% sucrose overnight. Tissues were embedded in OCT and stored at −80°C.

For the Oil Red O staining, 10-$\mu$m sections were fixed in 10% neutral buffered formalin, rinsed twice in isopropanol, and stained in freshly prepared Oil Red O working solution (3 g/60 ml isopropanol:40 ml distilled water). Slides were rinsed in isopropanol, counterstained with Mayer's haematoxylin (Sigma-Aldrich), and mounted using aqueous mounting medium H5501 (Vector Laboratories). Randomized images were obtained using a Zeiss Axio-Scan Z1 slide scanner. From each liver section, five smaller images were used for quantitative analysis using Fiji, as described in Graelmann et al (2024). Briefly, the intensity of the red signal was calculated and the optical density was estimated by normalizing the mean intensity to the maximal intensity.

### Estimation of lipocalin-2 levels

Faecal lipocalin-2 levels were measured in stool homogenates using the mouse lipocalin-2/NGAL DuoSet ELISA (R&D Systems) according to the manufacturer's instructions, and values were normalized to the weight of the faecal pellets.

### Cell isolation

Colon and small intestine were cleaned of luminal contents, cut open longitudinally, and washed in PBS. The epithelial layer was removed by incubation in HBSS without $Ca^{2+}/Mg^{2+}$ (Gibco), 5% FBS, and 2 mM EDTA (Invitrogen) for 40 min at 37°C at 200 rpm. Intestinal pieces and the c-MLN were washed in PBS, cut into small pieces, and digested with 1.5 mg/ml Collagenase VIII (Sigma-Aldrich), 50 $\mu$g/ml DNase I (Roche) with 5% FBS (Sigma-Aldrich), 15 mM Hepes (Gibco) in $Ca^{2+}/Mg^{2+}$-free HBSS for 30 min. Cells were filtered through a 100-$\mu$m cell strainer, washed in PBS with 5% FBS, filtered through a 70-$\mu$m strainer, and washed again.

### Flow cytometry

Cell suspensions were prepared as described for the indicated organ and incubated with anti-CD16/32 (eBioscience) and LIVE/DEAD Fixable Near-IR Dead Cell Stain Kit (Invitrogen) for 15 min at 4°C and washed in PBS. For surface stainings, cells were incubated with directly conjugated antibodies in PBS with 5% FBS at 4°C for 20 min. Cells were washed in PBS with 5% FBS and optionally fixed in 4% formaldehyde in PBS for 45 min to 18 h at 4°C. For intracellular staining of transcription factors or cytokines, cells were fixed and permeabilized with the Foxp3/ Transcription Factor Staining Buffer Set (eBioscience), according to the manufacturer's instructions. For intracellular stainings of cells obtained from AHR-TdTomato mice, cells were fixed and permeabilized using the Transcription Factor Phospho Buffer Set (RUO) (BD Biosciences), according to the supplied

protocol. Samples were acquired following standard procedures on BD FACSymphony A5 Cell Analyzer (BD Biosciences). Acquired data were analysed using FlowJo software (https://www.flowjo.com/).

B cells were gated as Live CD45+ Lin- (CD3, CD4, CD8, F4/80, CD11b, Gr1, Ter119) and analysed as follows: B1 (CD19+ B220−), B2 (CD19+ B220+), GL7+ CD38− B cells (GL7+ CD38− CD19+ B220+), IgD− GL7+ B cells (IgD− GL7+ CD19+ B220+). Myeloid cells were gated as Live CD45+ and defined as follows: dendritic cells (CD64− CD11c+ MHCII+), CD11b+ DCs (CD11b+ CD64− CD11c+ MHCII+), CD103+ DCs (CD103+ CD64− CD11c+ MHCII+), CD11b+ CD103+ DCs (CD11b+ CD103+ CD64− CD11c+ MHCII+), macrophages (CD64+ MHCII+), pDCs (CD11c$^{lo}$ MHCII− Siglec-H+ B220+). ILCs and T cells were gated as Live CD45+ CD90+ and analysed as follows: ILC (CD3−), T cells (CD3+), CD4+ T cells (CD4+), Foxp3+ CD4+ T cells (CD4+ Foxp3+), Rorgt+ CD4+ T cells (Rorgt+ CD4+), Tfh cells (PD-1+ CXCR5+), CD8+ T cells (CD8a+).

Experimental groups were masked during the preparation of the cell suspension and data analysis.

### Colon explant cultures and IL-22 ELISA

Colon tissue pieces (0.5–1 cm length) were weighed and cultured for 24 h at 37°C in complete IMDM. IL-22 cytokine levels in the supernatants were determined by ELISA (Invitrogen), and concentrations were normalized to the weight of the explants.

### RNA isolation, cDNA synthesis, and qRT–PCR

RNA was isolated from intestinal and white adipose tissue using TRIzol reagent (Invitrogen), according to the manufacturer's protocol. cDNA was synthesized with High-Capacity cDNA Reverse Transcription Kit (Applied Biosystems), and real-time quantitative PCR was performed using TaqMan Universal PCR Master Mix (Applied Biosystems) and respective probes. mRNA expression was determined using the ΔΔCt method, relative to hypoxanthine-guanine phosphoribosyl-transferase (Hprt) gene expression, and presented as fold change ($2^{-\Delta\Delta Ct}$).

### NP-Ficoll and NP-CGG immunizations and measurement of anti-NP antibodies

NP-Ficoll or NP-CGG (LGC Genomics) was dissolved in PBS and mixed with Alhydrogel (InvivoGen) at a 1:1 volume ratio. Mice were immunized with 10 µg NP-Ficoll or NP-CGG administered as a volume of 100 µl intraperitoneally. Fourteen days after immunization, whole blood was collected in Micro sample tube Serum Gel (Sarstedt) and serum was prepared by centrifugation at 10,000g for 5 min. For measuring anti-NP antibodies, plates were previously coated with 10 µg/ml NP-Ficoll or NP-CGG overnight at 4°C (Akkaya et al, 2017). Coated plates were washed in PBS plus 0.05% Tween-20, blocked in PBS plus 1% BSA, and washed before the addition of a fivefold serial dilution of sera. Samples were incubated for 2 h at RT. Ig isotypes were detected with goat anti-mouse IgG1, IgG2b, IgG2c, IgG3 antibodies conjugated to alkaline phosphatase

(SouthernBiotech) diluted in PBS + 1% BSA and incubated for 1 h at RT. After washing, plates were developed with p-nitrophenylphosphate (pNPP) substrate and read in a Tecan plate reader at OD$_{405}$.

### Anti-*C. rodentium* Ig ELISA

Analyses were performed on faecal lysates or serum prepared from whole blood collected in Micro sample Serum Gel tubes (Sarstedt) and by centrifugation at 10,000g for 5 min.

Heat-killed *C. rodentium* was prepared as previously described (Bry & Brenner, 2004). Briefly, an 18-h culture was resuspended in PBS plus a cOmplete protease inhibitor cocktail (Sigma-Aldrich) up to a volume with an OD$_{600}$ of 1.0. The culture was heat-killed by incubation at 60°C for 1 h, aliquoted, and frozen at −80°C until further use.

For the ELISA, aliquots were diluted 50 times in PBS for coating the plates, and stored overnight at 4°C. Anti-*C. rodentium* antibodies were detected using goat anti-mouse IgA, IgG1, IgG2b, IgG2c, IgM, and IgG3 antibodies conjugated to alkaline phosphatase (Southern Biotech), following the protocol described in the above section.

### Single-cell multiome sequencing of immune cells from the c-MLN and colon

Four to six male mice per group received an oral administration of vehicle or TCDD 6 d before infection with $2 \times 10^9$ CFU *C. rodentium* as described above. At day 5 post-infection, cells from the c-MLN and colon lamina propria were isolated as described above and pooled within the respective organ and group. The immune fractions were enriched using CD45 MicroBeads according to the manufacturer's protocol. Cells were stained with CD45, CD11c, MHCII, and CD64 surface antibodies and DRAQ7 Dye to exclude non-viable cells, and then sorted on a BD FACSAria Fusion into a 1.5-ml tube with PBS + 5% FCS using a 100-µm nozzle and a sorting precision protocol 16-16-0. Immune cells (Live CD45+) and dendritic cells (Live CD45+ CD11c+ MHCII+ CD64−) were sorted individually and mixed at a 1:1 ratio, for a total of 60,000–160,000 cells per sample. Cell nuclei were isolated according to the Demonstrated Protocol: Nuclei Isolation for Single Cell Multiome ATAC + Gene Expression Sequencing (CG00053 – Rev C, 10x Genomics) with the following modifications: nuclei for all samples were isolated using the Low Cell Input Nuclei Isolation protocol where cells were centrifuged at 300g for 5 min at 4°C and resuspended in 50 µl PBS + 0.04% BSA, lysed in chilled lysis buffer for 4 min on ice. After the incubation, wash buffer was added without mixing and the nuclei were centrifuged for 5 min at 500g at 4°C. Nuclei were washed once in Diluted Nuclei Buffer and resuspended in Diluted Nuclei Buffer. The concentration and quality of the single-nucleus suspension were measured using (1) acridine orange (AO) and propidium iodide (PI) and (2) Luna-FX7 Automated Cell Counter. Approximately, (3) 5,000–10,000 nuclei were transposed, then loaded on Chromium Chip, and partitioned in nanolitre-scale droplets using the Chromium Controller and Chromium (4) Next GEM Single Cell Reagents (Chromium Next GEM Single Cell Multiome ATAC + Gene

Expression Reagent Kits User Guide, User Guide, CG000338). A pool of 736,000 10x barcodes was sampled to separately and uniquely index the transposed DNA and cDNA of each individual nucleus. ATAC and gene expression (GEX) libraries were generated from the same pool of pre-amplified transposed DNA/cDNA and sequenced using the Illumina (5) NovaSeq 6000 (ATAC) and NovaSeq X (GEX). Sequencing read configuration is as follows: 28-10-10-90 (GEX), 50-8-24-49 (ATAC). The 10x barcodes in each library type are used to associate individual reads back to the individual partitions, and, thereby, to each individual nucleus. We aimed for 50-k reads per cell for GEX libraries, and 25-k reads for ATAC libraries.

### Statistics and data analysis

All statistical analyses were performed with GraphPad Prism software (https://www.graphpad.com). Description of the statistical tests used and the number of replicates are provided in the corresponding figure legends. Data points and n values represent biological replicates and are indicated in the respective figure legend.

### Quantification of gene expression and chromatin accessibility

Libraries were sequenced using Illumina NovaSeq 6000 or NovaSeq X platforms in accordance with 10x recommendations. Sequencing replicates for both modalities were provided to Cell Ranger ARC version 2.0.1 for quantification and filtering of GEMs for putative cells. The 10x-provided "refdata-cellranger-arc-mm10-2020-A-2.0.0" reference was used with default parameters otherwise.

### Filtering Cell Ranger ARC output

After initial QC of the Cell Ranger ARC output, the data were read into a Seurat object using Seurat (5.0.1) and Signac (1.14.0) to manage the gene expression and chromatin accessibility assays, respectively. All analysis used R version 4.4.1.

Putative cells were filtered according to the number of UMI and features reported per cell and the proportion of mitochondrial expression, selecting for genes on the mitochondrial chromosome in the index's GTF. Thresholds for these metrics were determined by visual inspection of the distributions in each dataset of each metric, with consideration given to the number of cells retained after filtering.

Doublets were predicted in each unfiltered dataset using the scDblFinder package (1.18.0). The MLN vehicle, AHR knockout, and TCDD datasets showed distinct clusters of doublets, which were removed. Similar clusters of putative doublets were not identified in the other datasets.

The filtering parameters used for each dataset, shown in Table S4, retained between 1,538 and 12,830 (mean 7,201) cells per dataset.

### Gene expression analyses

The gene expression assay was analysed using a Seurat pipeline: NormalizeData, FindVariableFeatures, ScaleData, RunPCA, RunUMAP. The number of PCs used for each dataset was determined by considering visual inspection of the elbow plot, marker gene loadings in principal components, and results of intrinsicDimension (1.2.0), and is provided in Table S5.

Clusters were defined using the FindNeighbors and FindClusters functions, with a range of resolutions from 0.2 to 2. Known marker genes were used to classify cell types independently for each dataset at an appropriate resolution, aided by clustree (0.5.1) analysis.

Experimental and control Seurat objects were merged into a new object while retaining the original cell-type labels. Differentially expressed genes were then identified using the RNA assay within cell types between conditions using the FindMarkers function of Seurat, with the Wilcoxon rank-sum test and correction using the Bonferroni method. The results were subsequently filtered to select features with minimum representation of 0.8 in the cell type, minimum absolute $\log_2$ fold change greater than 0.5, and adjusted $P$-value less than 1%. Functional enrichment was tested using the enricher function of clusterProfiler (4.12.6), within the biological processes defined by the msigdbr package (7.5.1), and a $P$-value cut-off of 1%. A summary table with the number of differentially expressed genes and the number of genes with higher expression in the experimental group is provided for each comparison in Table S6.

### Chromatin accessibility analyses

Peaks of transposition were identified within cell types using macs2 (2.2.7.1) implemented in Signac's CallPeaks function. Peaks were linked to TSS using the LinkPeaks function with default parameters.

Analysis of the chromatin accessibility within cell type–specific peaks followed the Signac pipeline: normalization by RunTFIDF, variable peak selection with FindTopFeatures, and scaling and decomposition using RunSVD. The "lsi" reduction was used to calculate t-SNE and UMAP reductions using dimensions 2–15 and the RunTSNE and RunUMAP functions. Clusters were identified using the FindNeighbors and FindClusters functions at a range of resolutions from 0.2 to 2.

Differential accessibility was assessed in unified peaks, defined using the UnifyPeaks function of Signac to implement the reduce function of GenomicRanges (1.56.1), within cell types between conditions using the FindMarkers function, as above. For comparison between chromatin and gene expression assays, the minimum representation in cell-type thresholds was relaxed to 0.2 and 0.05 for genes and peaks, respectively. Links between differentially expressed and differentially accessible features were identified from the results of LinkPeaks, with a maximum distance of 10 kb between the peak and TSS.

## Data Availability

Single-nucleus RNA-seq and ATAC-seq datasets produced in this study are deposited in the Gene Expression Omnibus (GEO) under the accession GSE297700. Supplementary AHR KO snMultiome datasets from both colon and c-MLN are available in the same GEO

submission but were not included in the article. Raw and processed data, including interactive Loupe Browser files, are provided for all datasets.

# Supplementary Information

# Acknowledgements

This work was supported by the Francis Crick Institute, which receives its core funding from Cancer Research UK, The UK Medical Research Council, and the Wellcome Trust (FC001159). It was further supported by Wellcome Trust Grant 210556/Z/18/Z to B Stockinger. E Wincent is supported by the Swedish Research Council VR (2020-03418). L Zhou is an Investigator in the Pathogenesis of Infectious Disease and is supported by the Burroughs Wellcome Fund. This work was supported by the National Institutes of Health grants AI132391 and AI157109 (to L Zhou). We would like to acknowledge the Science Technology Platforms at the Francis Crick Institute. We thank the Biological Research Facility for breeding and maintenance of our mouse strains, the Flow Cytometry Facility, the Advanced Sequencing Facility, and the Histopathology Facility.

## Author Contributions

OE Diaz: data curation, formal analysis, investigation, visualization, methodology, and writing—review and editing.
L Zhou: resources and methodology.
C Barrington: software, formal analysis, and methodology.
D Lindqvist: data curation, investigation, and methodology.
F Graelmann: data curation, formal analysis, and methodology.
E Wincent: conceptualization, data curation, supervision, investigation, methodology, and writing—original draft, review, and editing.
B Stockinger: conceptualization, supervision, funding acquisition, investigation, methodology, project administration, and writing—original draft, review, and editing.

## Conflict of Interest Statement

The authors declare that they have no conflict of interest.

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
