## [Reviewer comments · Life Science Alliance]

Beneficial and detrimental consequences of AHR activation in intestinal infection

Oscar Diaz, Liang zhou, Christopher Barrington, Dennis Lindquist, Frederike Graelmann, Emma Wincent, and Brigitta Stockinger

DOI: <https://doi.org/10.26508/lsa.202503414>

Corresponding author(s): Brigitta Stockinger, The Francis Crick Institute and Emma Wincent, Karolinska Institutet

Review Timeline:

Submission Date:	2025-06-09
Editorial Decision:	2025-07-18
Revision Received:	2025-08-12
Editorial Decision:	2025-09-09
Revision Received:	2025-09-11
Accepted:	2025-09-12

Scientific Editor: Sarita Hebbar

Transaction Report:

July 18, 2025

Re: Life Science Alliance manuscript #LSA-2025-03414-T

Dr. Brigitta Stockinger
The Francis Crick Institute
Mill Hill Laboratory
The Ridgeway
Mill Hill, London NW7 1AA
United Kingdom

Dear Dr. Stockinger,

Thank you for submitting your manuscript entitled "Beneficial and detrimental consequences of AHR activation in intestinal infection" to Life Science Alliance. The manuscript was assessed by three expert reviewers, whose comments are appended to this letter.

As you will read, all the reviewers were consistent that this work is potentially significant. That said, they also had several common concerns centred around the need for deeper data analyses and more details. Overall we concur with the reviewers that no new experimental data is required.

1) They pointed to both unsuitable statistical analyses and the lack thereof in some experiments. We concur with the reviewers that these aspects need to be addressed in line with their suggestions/concerns. 2) All the reviewers had concerns on the comparisons of the transcriptional response of the CYP gene. We agree with the reviewers that the authors should incorporate these suggestions to their multiomics analyses. 3) Reviewer 2 and 3 have raised concerns that the methods section is lacking pertinent information, the introduction section is not focussed, data interpretation and discussion sections are not fully supported by presented data. We agree that these sections need to be revisited in the revised manuscript and carefully edited taking into account of the reviewers' suggestions.

We invite you to submit a revised manuscript addressing the reviewers' comments. When submitting the revision, please include a letter addressing the reviewers' comments point by point. While a rebuttal must respond to all points in some form, additional experiments to resolve these points is not required.

Thank you for this interesting contribution to Life Science Alliance. We are looking forward to receiving your revised manuscript.

Sincerely,

Sarita Hebbar, PhD
Scientific Editor
Life Science Alliance
<http://www.lsajournal.org>

B. MANUSCRIPT ORGANIZATION AND FORMATTING:

Reviewer #1 (Comments to the Authors (Required)):

This is an important report from an outstanding team that has made numerous contributions to the fields of Ah receptor (AHR) biology/immunology. Understanding the role of the AHR in vertebrate health has been an objective for almost 75 years, with the prototype ligand, 2,3,7,8-tetrachlorodibenzo-p-dioxin (TCDD) serving as the prototype agonist. The more recent development of genetic models of signaling, such as the CAIR line employed here is providing a new level of understanding of signaling in a model organism of human toxicology.

The experimental results reported are a complex set of both single cell gene expression and chromatin accessibility data for the GI tract from both TCDD exposed and CAIR mice. This is coupled with some fundamental toxicological endpoints (e.g., thymic weight and oil-red staining of liver). This is important data and of considerable value to the AHR community. The described distinctions between these two data sets (TCDD and CAIR) are robust and appear significant. The experiments are carefully described and the data analyzed with the methods of the day. In addition, the deposition of the associated data for use by the larger scientific community will provide an important resource for many biologists in the coming years.

My main criticism has to do with the interpretation of results. Essentially, the authors conclude "Thus, the detrimental impact of environmental pollutants such as TCDD on immune responses cannot solely be attributed to aberrantly prolonged activation of AHR." I would recommend a statement more along the lines of: "Activation of the AHR via a high dose of a recalcitrant, high potency ligand, is not equivalent to the response from a constitutively active AHR. It may be a misconception to attribute TCDD toxicity to prolonged activation, it may be more of an "area under the curve" phenomenon, both temporal and magnitude of response.

The discussion could benefit from the consideration of the following:

- 1) CAIR mice have experienced activation of the AHR throughout life, the TCDD mice only experienced this activation for about 6 days. It is interesting that the thymic involution is so similar under these two distinct conditions. While it would be cost prohibitive to perform more time points or TCDD doses, this limitation should be pointed out to the reader.
- 2) The comparison of the transcriptional responses of CYP genes to TCDD and CAIR looks quite similar in Log2 space. If this is replotted in base 10, the difference might be about 8-fold? The therapeutic index of acetaminophen is ~ 10-fold, digoxin is 2. The two AHR activation conditions may not be as comparable as the robust conclusion would depend.

Cross-review comments:

I don't have any significant disagreements with either reviewer. In fact, I think we picked up on many of the same issues. Although, I was a bit more positive about the manuscript.

Reviewer #2 (Comments to the Authors (Required)):

This paper presents a set of experimental findings anchored in trying to solve puzzle: why does influencing AHR signaling have similar or dissimilar consequences-in this case to the immune system? This is an important question with broad-reaching

implications to the fields of pharmacology, toxicology and immunology. The paper reports findings from several different approaches, with the main approach being parallel comparisons in mice given the AHR ligand TCDD and mice that constitutively express AHR (AhrdCAIR/dCAIR mice). A major strength of the work reported in this paper is the use of in vivo model systems to examine how differences in AHR signaling contribute to a mixture of similar and dissimilar consequences. Overall, some of the findings are novel, while others align well with prior studies, and provide solid contextual framework for the models used, but are not novel. A concern is that the totality of the work presented in this paper seems disconnected, both regarding rationale and conclusion. That is, the results in the paper seem like disparate assessments that were stitched together, often without a strong rationale for specific assessments presented. In some instances, the results reported, at least based on the information provided and analyses performed, do not robustly support some of the conclusions.

Major concerns:

1. While the work appears to have been conducted well, and overall the findings are consistent with the gist of the conclusions, there are some instances in which the rationale for specific assessments is lean or based on poorly substantiated reasoning, and some of the conclusions are not strongly supported by the work presented. Three examples are provided here:
 - The paper contains an analysis of scRNA seq and ATAC seq data focusing on dendritic cells (DC), but none of the prior results presented in the paper, or summarized in the introduction, provide a strong rationale for narrowing in on DC.
 - Comparing Cyp1a1 levels in various tissues in mice treated with TCDD and I3C is not illogical, but also not particularly novel- and how this relates to the comparison of AhrdCAIR/dCAIR mice and TCDD treated mice is unexplained.
 - The limited comparison and lean assessments in mice infected with *C. rodentium* and treated with DSS to induce intestinal inflammation are not sufficient with regard to experimental design or 'depth' of analyses to support conclusions about cell type specificity.
2. The authors are encouraged to revisit the statistical analyses. In some of the figures, the p-values reported and the range of datapoints within groups are difficult to reconcile. That is, a lot of the data have large error bars, yet the authors report p-values that are sometimes denoted as <0.001 . It is challenging to understand how this is possible, especially given the relatively small 'n' in each group. Another concern is that the authors mention identifying and removing statistical outliers, but provide no information on the number of samples, or a strong justification for removal of samples. It is potentially problematic to remove a sample from one plot but retain data from that mouse in another plot. Insufficient information is provided to appreciate whether these points represent major or minor concerns.
3. The paper is not particularly well prepared. For example, uncited and unsubstantiated information is presented throughout the paper, particularly in the Introduction and Discussion. Several examples are provided:
 - a. In some instances, findings that are not novel are presented in a manner that inadvertently conveys novelty. For example, prior research studies have shown that AHR activation using TCDD elicits toxicity, such as to the liver and thymus, and that hepatotoxicity and thymic atrophy are not linked to immunomodulatory action of TCDD (or other AHR ligands). Yet, in this paper the authors present these findings as if they are new information.
 - b. Some results are presented for which a rationale or connection to the overall hypothesis or to the prior findings presented in the paper is absent or very lean (e.g., comparing Cyp1a1 expression in mice treated with TCDD vs I3C, analysis of gene expression and chromosome accessibility analyses in DCs).
 - c. Lean citations of prior work pervade the paper, including the introduction, and many statements are poorly or unsubstantiated. An example is the opening sentence of the Results section; however, any single example in isolation is minor. The concern is that this issue is peppered throughout the paper.
 - d. Methods reporting contains some gaps in information, such as source of mice and how the authors address sex as a biological variable (e.g., they need to be much more transparent about the number of male and number of female mice in each group for all conditions, and whether animal sex could be a factor in differences observed-or in some of the large error bars). In assessments such as gene expression and chromatin accessibility, sex is likely an important variable. More detail regarding analyses of gene expression and chromosome accessibility data sets is necessary.
4. Experimental rationale and robustness need some clarification. As already mentioned, the authors need to indicate whether all data presented were derived from male or female mice, or a combination. This could be added to each figure legend, denoting, for example, the total group size within each group that includes the number of male and number of female mice. In the absence of this information, it is difficult to appreciate whether some of the variation within and across groups is due to sex or experimental conditions. Additional aspects of the rationale that would benefit from further clarification or for which it is difficult to appreciate how the experimental work aligns with the stated goal of the study:
 - a. Comparing constitutively active AHR mice with mice treated with TCDD makes sense and is novel; however, to limitations of these two approaches need some consideration this comparison does not rule out that there are not additional factors, in addition to duration that are main driver of toxicity.
 - b. The authors measured TCDD levels in organs, to provide a metric of internal dose. This represents substantive effort; however, studies in the 1980s and 1990s analyzed TCDD ADME in rodents quite extensively and established that the half-life of TCDD in mice is roughly 7-10 days, depending on strain. Therefore, the purpose of measuring TCDD levels in the time frame presented is unclear.
 - c. The rationale for focusing on DCs was justified in the broad context of prior studies showing that AHR signaling impacts DCs (although citation of these prior studies is inaccurate). However, there is not a clearly presented rationale for connecting the goal of this study (and findings in Fig 1-4) to the deep dive into gene expression and chromatin accessibility in DCs. A minor point:

DCs do not have 'exceptionally high AHR expression.' What is considered high and low is relative, and it is not entirely clear that relative abundance of AHR in a particular cell type is predictive of sensitivity to AHR modulation.

d. Assessing AHR levels using TdTomato expression reporter is creative. However, the data presented have no error bars, and n of 2 per group is insufficient for statistically robust analyses. The histograms are presented in extended materials, and this is helpful. However, the absence of gating strategy and the absence of hypothesis make these data difficult to contextualize (e.g., how does the level of AHR expression relate to answering the question of whether immunomodulatory differences are due to prolonged AHR activity?). To help readers better appreciate this analysis of AHR expression it would also be helpful to present negative controls, gating information, as well as percentage data.

e. While I appreciate that results sometimes contain a lot of variability, at times the level of variability between control groups (e.g., Fig 3, when comparing WT and vehicle control groups at the same day after infection, there appears to be considerably variability in CFU, IL-22 and fecal Lcn2 levels). Similarly, in Figure 4 there are wide and often overlapping error bars, yet indications that means are significantly different (statistically speaking).

f. Some of the main conclusions are based on differences between AhrdCAIR/dCAIR mice and TCDD treated mice. However, there are no statistically rigorous approaches used to compare endpoints in these two groups. Perhaps this reflects aspects of the study design, but the authors need to be much more circumspect in the conclusions drawn because they did not directly compare these two conditions. Absent this, a lot of their conclusions are not strongly supported. One example are the findings in Figure 3.

Reviewer #3 (Comments to the Authors (Required)):

Summary:

In this study, the authors investigate whether prolonged activation of the aryl hydrocarbon receptor (AHR) is the primary driver of impaired host immune responses during infection. To explore this hypothesis, two distinct models of chronic AHR activation were examined: a genetic model (constitutive AHR activation) and ligand activation of the AHR with 10 µg/kg TCDD administered once. The results suggest prolonged activation via TCDD impairs bacterial clearance, which is associated with reduced antibody production. Moreover, despite the claim that genetic constitutive AHR activation enhanced infection clearance, the presented data showed little to no effect. Using integrated single-cell RNA-seq and ATAC-seq analyses, evidence is presented that TCDD, but not constitutive genetic activation, compromised key dendritic cell functions, including activation, maturation, and antigen presentation. This work offers novel insight into the differential effects of AHR activation on immune function. However, several concerns need to be addressed.

Comments:

Comparison of CAIR and TCDD Models

-In Figure 2, the level of Cyp1a1 induction in the colon does not appear comparable between constitutively active AHR (CAIR) mice and TCDD-treated mice. Please report actual fold change presented in A and B. Could the difference in the level of induction in Cyp1a1 suggest AHR activation was not equivalent? Was a similar trend seen with other known AHR responsive genes (Ahrr, Tiparp, Cyp1a2, Nqo1)? The conclusion that persistent activation may not solely explain the effects on the immune responses may need qualification (such as testing for whether there is a difference in Cyp1a1 expression in the colon between CAIR mice and TCDD-exposed wild-types) since the potency of gene expression in CAIR mice may not be equivalent to the level of gene expression induced by 10 µg/kg TCDD.

-The lack of clearance of *C. rodentium* from TCDD-exposed mice relative to the wild-type controls and the CAIR mice is convincing; however, the case for a difference between the TCDD-exposed and CAIR mice would be strengthened if they were directly compared with statistics. Moreover, a direct statistical comparison between the TCDD-treated and constitutively active AHR models should be included to support the conclusion about IL-22 levels and Lipocalin-2.

-The most salient point presented is that TCDD appears to have suppressed infection clearance, while constitutively activated AHR did not. Accompanying this observation, that single nuclear and ATAC-seq data of colon-associated immune cells, identify putative differences between TCDD treated and constitutive AHR activation. Yet, the multiomic analysis is underexplored. A direct comparison between the TCDD treated and constitutive AHR models is needed, in addition to what is present, beyond the indirect comparison between the two conditions by comparing the results of the comparison of each to the vehicle.

-TCDD suppressing dendritic cell function is interesting, but possible underlying mechanisms are not explored. Further analysis of dendritic gene expression by GSEA (or equivalent) would strengthen the conclusions and demonstrate the relevance of the finding provided the number of dendritic cells sequenced in the single-cell RNA-seq assay were sufficient.

-Were AHR protein levels assessed in TCDD-treated vs. CA-AhR animals? Given the negative feedback on AHR expression, reduced receptor levels might explain the lower Cyp1a1 expression in the CA-AhR model.

-In EV1, fecal Lcn2 levels differ between WT and vehicle groups. This is unexpected and should be discussed. Could these differences influence the infection dynamics observed between TCDD- and CA-AhR-treated mice?

-In EV1C and EV1F, more timepoints are shown for vehicle and TCDD groups compared to WT and CA-AhR. Timepoints should be consistent across all groups to support meaningful comparisons.

Data Interpretation Issues

-The statement, "AhrdCAIR/dCAIR mice cleared the bacteria faster than wildtype mice (Fig.3A)" was not supported by the CFU/g faeces barplot from Fig. 3A, especially considering the unequal biological replicates between wild-type and CAIR mice.

-The authors stated that "AHR expression levels were higher in cells from the colon and the highest expression was seen in the myeloid lineage comprising dendritic cells, macrophages, and eosinophils". What statistical test was performed to prove that

statement? Please provide SD bars in Figure 1.

-In the "Methods" subsection "Statistics and data analysis", the authors state: "Statistically significant outliers were identified using the ROUT method (Q = 5%) and removed from the plots." This rule of thumb is alarming considering the potential abuse it permits, including the removal of biological variation because of a perceived technical error.

-The statement that TCDD levels changed across tissues over 6 days needs statistical analysis (results; Fig. 2E). Please include appropriate tests to show that these differences are significant.

-The authors state that "Fig. 2F-H clearly shows that both models of prolonged AHR signaling exhibited clear toxic endpoints," including intrahepatic lipid accumulation. However, in Fig. 2H, there is no visible effect of TCDD or AhrdCAIR/dCAIR on lipid accumulation, as presented in the graph on the right. The authors should either include statistical analysis to support this claim or avoid drawing conclusions based solely on the visual appearance.

Experimental Design Clarity and Control Group Consistency

-In the "Mice" subsection of "Methods and Protocols", the strain of the wild-type mice is reported to be C57BL/6J and a reference to the Jackson laboratory is made in the "Reagents and Tools table". However, the background of the AhrCAIR strain is unclear from the manuscript alone, and the reference for the strain generation was also ambiguous. When directly comparing differences between TCDD or vehicle-treated wild-type and non-exposed CAIR mice (e.g., Fig. 2F - H), the "wild-type" mice should have the same genetic background as the mice with CAIR alleles. Otherwise, differences in strain background could explain differences between the CAIR and TCDD treated models as well as the non-exposed wild-type mice (e.g. Fig. 3A).

-The authors state that they used either male or female mice, but their analyses do not distinguish results by sex, suggesting that they lumped together. Further clarification about the sex of the mice in this study is required.

-The study design of the single-nuclear RNA-seq and ATAC-seq experiment needs clarification. For example, the first sentence of the subsection "Single-cell multiome sequencing of immune cells from the c-MLN and colon" in the "Methods" is garbled: "Four to six mice per group received an oral administration of with vehicle TCDD 6 days prior to infection with 2×10^9 CFU C. rodentium as described above." This reader assumes there was a vehicle group to which the TCDD treated and CAIR mice were compared.

-A visual summary of the experimental setup would enhance the clarity and reader understanding.

Single-Cell / Multiomic Analysis and Methods Concerns

-The authors performed unsupervised clustering in part determined using the FindMarkers method from Seurat. FindMarkers is not a reliable source for identifying DEGs because the p-values are inflated from treating cells as replicates and from "double-dipping" (i.e., using the same gene expression data that was used to cluster cells into cell types to calculate p-values for differentially expressed clustered- or cell-type specific genes). An alternative to DEG overrepresentation, such as gene set enrichment analysis (GSEA), would avoid the calculation of p-values. If the authors pursue an alternative method to calculate DEGs, they should consider reporting basic information on the DEGs (e.g., the number of DEGs per cell type and condition).

Data Presentation and Figure Labeling Issues

-Please provide the TCDD levels reported in Fig 2E in a supplementary table. A brief comparison to previous studies using the same or similar designs in which TCDD was administered and measured in the same tissues should be added to corroborate the GC-MS method.

-Figure 5C is a difficult read. The gene ontologies should be identified by their gene ontology code in the figure legend or in the manuscript text to enhance reproducibility.

-Figures 5 and EV5 contain panels labeled "CAIR," which is inconsistent with the rest of the manuscript where the model is referred to as AhrdCAIR/dCAIR. Additionally, "CAIR" is mentioned in the EV1 legend-please clarify whether this refers to AhrdCAIR/dCAIR or correct the labeling if it is an error.

-The distinction between panels A and B in EV2 is unclear. The figure legend implies they show the same data. Please revise the legend or clarify the intent of each panel.

Methodological Details / Reagents and Protocols

-For the subsection "NP-Ficoll and NP-CGG immunizations and measurement of anti-NP antibodies" in "Methods", a brief explanation for why multiple serum dilutions were used is requested to provide context, especially for readers without a background in immunology.

-Cell isolation: What is the rationale of adding FBS to digestion buffer? It is well known that FBS inhibits enzymatic activity of collagenase, and it is typically added when resuspending the cells after digestion to stop collagenase activity.

Other

-The supplementary appendix was not included and thus could not be evaluated. Please ensure all supplementary files are provided for peer review.

Response to Reviewers:

Reviewer 1

My main criticism has to do with the interpretation of results. Essentially, the authors conclude "Thus, the detrimental impact of environmental pollutants such as TCDD on immune responses cannot solely be attributed to aberrantly prolonged activation of AHR." I would recommend a statement more along the lines of: "Activation of the AHR via a high dose of a recalcitrant, high potency ligand, is not equivalent to the response from a constitutively active AHR. It may be a misconception to attribute TCDD toxicity to prolonged activation, it may be more of an "area under the curve" phenomenon, both temporal and magnitude of response.

The discussion could benefit from the consideration of the following:

1) CAIR mice have experienced activation of the AHR throughout life, the TCDD mice only experienced this activation for about 6 days. It is interesting that the thymic involution is so similar under these two distinct conditions. While it would be cost prohibitive to perform more time points or TCDD doses, this limitation should be pointed out to the reader.

2) The comparison of the transcriptional responses of CYP genes to TCDD and CAIR looks quite similar in Log2 space. If this is replotted in base 10, the difference might be about 8-fold? The therapeutic index of acetaminophen is ~ 10-fold, digoxin is 2. The two AHR activation conditions may not be as comparable as the robust conclusion would depend.

We are grateful to the Reviewer for the positive comments on our manuscript. Regarding the two points of criticism, we have taken these points on board and modified the manuscript text to reflect these limitations of our study. In response to the other reviewers, we have changed the figure of Cyp1a1 expression comparison to show fold changes, but we are also providing a figure for this Reviewer in which the data are replotted in base10.

The new Fig.2A,B also shows induction of AHRR and now represents pooled data from 3 and 2 experiments, respectively. This does indeed show that the amplitude of gene activation in the $Ahr^{dCAIR/dCAIR}$ mice is lower than that seen after TCDD application. We would like to point out that our intention was not to claim that persistent activation in $Ahr^{dCAIR/dCAIR}$ mice is equivalent to that seen after TCDD, but to illustrate that prolonged activation compared to transient activation by physiological ligands does not necessarily equate to detrimental effects on immune responses.

Reviewer 2

Major concerns:

1. While the work appears to have been conducted well, and overall the findings are consistent with the gist of the conclusions, there are some instances in which the rationale for specific assessments is lean or based on poorly substantiated reasoning, and some of the conclusions are not strongly supported by the work presented. Three examples are provided here:

- The paper contains an analysis of scRNA seq and ATAC seq data focusing on dendritic cells (DC), but none of the prior results presented in the paper, or summarized in the introduction, provide a strong rationale for narrowing in on DC.*

The primary rationale for our focus on dendritic cells is the fact that this cell type underlies the activation of adaptive immune cells such as B cells and T cells and it therefore stands to reason that any compromise of their function will impact adaptive immune responses that shape the response to infection. We felt that the deep dive into single cell RNAseq and chromatin accessibility was essential to get beyond observations into potential mechanisms.

- Comparing Cyp1a1 levels in various tissues in mice treated with TCDD and I3C is not illogical, but also not particularly novel-and how this relates to the comparison of AhrdCAIR/dCAIR mice and TCDD treated mice is unexplained.*

We agree that this should have been expressed more clearly, and we have now included a sentence to explain the rationale of this comparison which was to have an internal comparison of AHR activation by the persistent TCDD ligand compared with a physiological ligand of similar affinity which however is rapidly cleared.

- The limited comparison and lean assessments in mice infected with C. rodentium and treated with DSS to induce intestinal inflammation are not sufficient with regard to experimental design or 'depth' of analyses to support conclusions about cell type specificity.*

We have included more detailed rationale in the manuscript to justify our focus on dendritic cells, but we have not made conclusions about exclusive cell type specificity. As the infection model with C.rodentium has been characterised in detail by us and others we have not – for the purpose of this manuscript- focused on extensive characterisation of immune populations affected by this infection but rather concentrated on the salient points that determine the response such as clearance of bacteria, production of IL-22 and antibodies. However, we agree with the reviewer that the analysis of the DSS response was not as extensive as needed to justify the conclusion that TCDD does not affect epithelium. We had added this to supplementary information as it goes back to previous findings on the role of AHR in epithelial cell differentiation, but it is insufficiently supported in our current manuscript. We have therefore removed these data and references to DSS.

2. The authors are encouraged to revisit the statistical analyses. In some of the figures, the p-values reported and the range of datapoints within groups are difficult to reconcile. That is, a lot of the data have large error bars, yet the authors report p-values that are sometimes denoted as <0.001. It is challenging to understand how this is possible, especially given the relatively small 'n' in each group. Another concern is that the authors mention identifying and removing statistical outliers, but provide no information on the number of samples, or a strong justification for removal of samples. It is potentially problematic to remove a sample from one plot but retain data from that mouse in another plot. Insufficient information is provided to appreciate whether these points represent major or minor concerns.

We appreciate this point and have now included all datapoints that were previously removed to be more transparent and adjusted the Methods section accordingly.

3. The paper is not particularly well prepared. For example, uncited and unsubstantiated information is presented throughout the paper, particularly in the Introduction and Discussion. Several examples are provided:

a. In some instances, findings that are not novel are presented in a manner that inadvertently conveys novelty. For example, prior research studies have shown that AHR activation using TCDD elicits toxicity, such as to the liver and thymus, and that hepatotoxicity and thymic atrophy are not linked to immunomodulatory action of TCDD (or other AHR ligands). Yet, in this paper the authors present these findings as if they are new information.

We are surprised that the reviewer thinks we have presented data on thymus and liver toxicity as novel. The sentence in the Results says: "TCDD toxicity is associated with hepatomegaly, intrahepatic lipid accumulation and thymic involution (Poland & Knutson, 1982)" and thus clearly indicates that this is a known phenomenon. In the Introduction we state "...both scenarios of prolonged AHR activation produced the known toxic effects on liver and thymus" so again we made it clear that this is a known phenomenon.

Nevertheless, toxicity has not previously been assessed for *Ahr*^{dCAIR/dCAIR} and we would have been amiss had we not checked this critical feature of aberrant AHR activation.

b. Some results are presented for which a rationale or connection to the overall hypothesis or to the prior findings presented in the paper is absent or very lean (e.g., comparing Cyp1a1 expression in mice treated with TCDD vs I3C, analysis of gene expression and chromosome accessibility analyses in DCs).

We agree that this should have been clearer, and have included sentences in both sections explaining the motivation for both analyses.

c. Lean citations of prior work pervade the paper, including the introduction, and many statements are poorly or unsubstantiated. An example is the opening sentence of the Results section; however, any single example in isolation is minor. The concern is that this issue is peppered throughout the paper.

We regret that the reviewer feels we have been deficient in citing previous work and used unsubstantiated statements, but in the absence of pointing out these omissions we find it difficult to correct this as we have not deliberately omitted citation of other researchers' work. The only example given refers to the first

sentence in the Results section which says: “AHR is widely expressed in the immune system in a cell type and context specific manner”.

Our thinking was that this is such a long-established fact that it would not need a citation and in fact it would be difficult to decide which work to cite. We have now included a reference from a review which will include a host of primary references on this topic.

d. Methods reporting contains some gaps in information, such as source of mice and how the authors address sex as a biological variable (e.g., they need to be much more transparent about the number of male and number of female mice in each group for all conditions, and whether animal sex could be a factor in differences observed-or in some of the large error bars). In assessments such as gene expression and chromatin accessibility, sex is likely an important variable. More detail regarding analyses of gene expression and chromosome accessibility data sets is necessary.

We state in the Methods that: “AHR-tdTomato, PGK-Cre, *Ahr*^{CAIR}, *Ahr*^{dCAIR/dCAIR} and C57BL/6J mice used in this study were bred and maintained in individually ventilated cages under specific pathogen-free conditions at the Francis Crick Institute, according to protocols approved by the UK Home Office and the Ethics committee of the Francis Crick Institute”.

We agree with the reviewer that sex as a biological variable in response to infection is very important and we should have been clearer in stating the composition of our groups. We checked that the clearance of infection was equivalent between males and females (now shown in Fig.S3) and therefore proceeded to use both sexes in our experiments, but we have now added clear information on the sex used in our experimental groups in the legends of all figures.

4. Experimental rationale and robustness need some clarification. As already mentioned, the authors need to indicate whether all data presented were derived from male or female mice, or a combination. This could be added to each figure legend, denoting, for example, the total group size within each group that includes the number of male and number of female mice. In the absence of this information, it is difficult to appreciate whether some of the variation within and across groups is due to sex or experimental conditions. Additional aspects of the rationale that would benefit from further clarification or for which it is difficult to appreciate how the experimental work aligns with the stated goal of the study:

As stated above we have now added detailed information on the sex of our experimental groups into the legends.

a. Comparing constitutively active AHR mice with mice treated with TCDD makes sense and is novel; however, to limitations of these two approaches need some consideration this comparison does not rule out that there are not additional factors, in addition to duration that are main driver of toxicity.

We fully agree with the reviewer that additional factors apart from duration of signalling must drive toxicity and we state this in the Discussion: “Taken together our data confirm previous data in the literature showing that TCDD results in

suppression of immune responses. However, they also clearly indicate that prolonged AHR activation **alone** cannot account for this...”

b. The authors measured TCDD levels in organs, to provide a metric of internal dose. This represents substantive effort; however, studies in the 1980s and 1990s analyzed TCDD ADME in rodents quite extensively and established that the half-life of TCDD in mice is roughly 7-10 days, depending on strain. Therefore, the purpose of measuring TCDD levels in the time frame presented is unclear.

We agree with the reviewer that the TCDD ADME has been thoroughly reported. Our aim was however not to determine the half-life of TCDD, for which a longer time period would have been needed, but to illustrate its local (IS/colon) and systemic (serum/liver/adipose) tissue distribution and thereby widespread AHR activation. We have added a sentence in the Result section to make this clearer.

c. The rationale for focusing on DCs was justified in the broad context of prior studies showing that AHR signaling impacts DCs (although citation of these prior studies is inaccurate). However, there is not a clearly presented rationale for connecting the goal of this study (and findings in Fig 1-4) to the deep dive into gene expression and chromatin accessibility in DCs. A minor point: DCs do not have 'exceptionally high AHR expression.' What is considered high and low is relative, and it is not entirely clear that relative abundance of AHR in a particular cell type is predictive of sensitivity to AHR modulation.

c) The primary rationale for our focus on dendritic cells is the fact that this cell type underlies the activation of adaptive immune cells such as B cells and T cells and it therefore stands to reason that any compromise of their function will impact adaptive immune responses that shape the response to infection. We felt that the deep dive into single cell RNAseq and chromatin accessibility was essential to get beyond observations into potential mechanisms. With respect to AHR expression levels on dendritic cells we concur that expression does not equal activation. Nevertheless, the analysis using reporter mice that allows assessment of protein levels rather than just mRNA levels on different cells types unequivocally shows that myeloid cells such as dendritic cells express considerably higher protein levels of AHR and are therefore very well equipped for activation compared with cell types that have very low expression. This has been corroborated in a previous publication (Diny et al 2022). Fig.1B showing % of positive cells also highlights that dendritic cells always express high levels of AHR irrespective of their location in gut vs lymphnode whereas other immune cell types, notably T cells are barely positive outside the intestinal environment.

It is not clear to us what was inaccurate about the citation regarding AHR signalling in DC, but we added an additional reference (Jin et al) which refers to AHR effects via TCDD on dendritic cells.

d. Assessing AHR levels using TdTomato expression reporter is creative. However, the data presented have no error bars, and n of 2 per group is insufficient for statistically robust analyses. The histograms are presented in extended materials, and this is helpful. However, the absence of gating strategy and the absence of hypothesis make these data difficult to contextualize (e.g., how does the level of AHR expression relate to answering the question of whether immunomodulatory differences are due to prolonged AHR

activity ?). To help readers better appreciate this analysis of AHR expression It would also be helpful to present negative controls, gating information, as well as percentage data.

The AHR-Tomato feature is a stable genetic modification and therefore not prone to variation in steady state like other measurements that require experimental manipulation. This is clearly indicated in the tight alignment of each of the two datapoints. Again, it would have been difficult to increase the group size as the digestion protocols for isolation of some of the rarer cell types are sensitive and delays likely resulting in reduced cell viability.

We have modified Fig.1 to show not only MFI for Tomato but also percentage values.

Furthermore, we have included the gating strategy in Fig.S2.

e. While I appreciate that results sometimes contain a lot of variability, at times the level of variability between control groups (e.g., Fig 3, when comparing WT and vehicle control groups at the same day after infection, there appears to be considerably variability in CFU, IL-22 and fecal Lcn2 levels). Similarly, in Figure 4 there are wide and often overlapping error bars, yet indications that means are significantly different (statistically speaking).

We agree with the reviewer that some of the data show considerable variability whilst still showing statistical significance. As the data in Fig.3 were all obtained with male mice it is unlikely that the reason for variability was a mixture of sexes. We have changed the display for Fig.4 to bar plots to make it more readable as the antibody response to Citrobacter was quite low overall.

f. Some of the main conclusions are based on differences between AhrdCAIR/dCAIR mice and TCDD treated mice. However, there are no statistically rigorous approaches used to compare endpoints in these two groups. Perhaps this reflects aspects of the study design, but the authors need to be much more circumspect in the conclusions drawn because they did not directly compare these two conditions. Absent this, a lot of their conclusions are not strongly supported. One example are the findings in Figure 3.

We appreciate this point and while this would have been the ideal approach, performing the experiments and readouts shown in Figures 3 and 4 for all conditions would have compromised cell viability for flow cytometry experiments.

However, the single cell omics experiments shown in Figures 5 and 6 were done with all 3 groups in parallel and are therefore directly comparable.

Comments:

Comparison of CAIR and TCDD Models

-In Figure 2, the level of Cyp1a1 induction in the colon does not appear comparable between constitutively active AHR (CAIR) mice and TCDD-treated mice. Please report actual fold change presented in A and B. Could the difference in the level of induction in Cyp1a1 suggest AHR activation was not equivalent? Was a similar trend seen with other known AHR responsive genes (Ahrr, Tiparp, Cyp1a2, Nqo1)? The conclusion that persistent activation may not solely explain the effects on the immune responses may need qualification (such as testing for whether there is a difference in Cyp1a1 expression in the colon between CAIR mice and TCDD-exposed wild-types) since the potency of gene expression in CAIR mice may not be equivalent to the level of gene expression induced by 10 µg/kg TCDD.

We have changed Fig2.A-C to now show actual fold change in panels A and B and also included measurement of another AHR target gene, AHRR. The new Fig.2A-B represents a pool of 3 and 2 different experiments, respectively, rather than a single experiment as previously shown.

It is correct that the amplitude of AHR activation exemplified by expression of Cyp1a1 is lower in AhR^{dCAIR/dCAIR} mice than what is seen after TCDD. However, the main point is the substantially prolonged AHR activity compared to physiological AHR activation. We consider it unlikely that the amplitude of activation should be the only factor responsible for the different functional outcomes of exposure to TCDD vs constitutive activity. It was not our intention to claim that the potency of gene expression in AhR^{dCAIR/dCAIR} mice is equivalent to that of TCDD, but rather to illustrate that substantially prolonged activity compared to physiological short lived AHR activation need not necessarily be detrimental for immune responses. Thus, whatever dictates the immune toxicity of TCDD is unlikely due to just prolonged AHR activation. We have now included a sentence in the discussion to make this clearer.

-The lack of clearance of C. rodentium from TCDD-exposed mice relative to the wild-type controls and the CAIR mice is convincing; however, the case for a difference between the TCDD-exposed and CAIR mice would be strengthened if they were directly compared with statistics. Moreover, a direct statistical comparison between the TCDD-treated and constitutively active AHR models should be included to support the conclusion about IL-22 levels and Lipocalin-2.

We agree that this is a valid point but unfortunately the experiments with AhR^{dCAIR/dCAIR} mice and TCDD were not done at the same time. This was due to technical reasons as it is difficult to handle too many gut preps at the same time due to rapid degradation but with hindsight we should have split the groups to always have mice of both treatment groups in one experiment. However, we can say that these differences repeated over several experiments despite the high variability of IL-22 and Lipocalin-2 levels.

-The most salient point presented is that TCDD appears to have suppressed infection clearance, while constitutively activated AHR did not. Accompanying this observation, that single nuclear and ATAC-seq data of colon-associated immune cells, identify putative differences between TCDD treated and constitutive AHR activation. Yet, the multiomic analysis is underexplored. A direct comparison between the TCDD treated and

constitutive AHR models is needed, in addition to what is present, beyond the indirect comparison between the two conditions by comparing the results of the comparison of each to the vehicle.

Our rationale for relating comparisons of CAIR and TCDD with the vehicle control was that we are arguing that prolonged activation per se is not detrimental to immune responses and for this the comparison with the vehicle group is relevant.

However, we have now provided gene ontology plots in Fig. S10 comparing both groups directly, not respective to Vehicle. This shows an enhancement of antigen presentation in dendritic cells from AhR^{dCAIR/dCAIR} mice in the c-MLN. The data shown in Figure 6 correspond to a direct comparison between TCDD-treated and AhR^{dCAIR/dCAIR} mice.

-TCDD suppressing dendritic cell function is interesting, but possible underlying mechanisms are not explored. Further analysis of dendritic gene expression by GSEA (or equivalent) would strengthen the conclusions and demonstrate the relevance of the finding provided the number of dendritic cells sequenced in the single-cell RNA-seq assay were sufficient.

We have applied GSEA using the clusterProfiler's gseGO function to cell types using the fold change reported by Seurat's FoldChange function. The results are presented in an additional figure to this document, where we observed that a larger fraction of genes involved in antigen processing and presentation pathways showed higher expression in dendritic cells from AhR^{dCAIR/dCAIR} mice compared to respective cells from either Vehicle or TCDD treated mice, supporting our conclusions.

-Were AHR protein levels assessed in TCDD-treated vs. CA-AhR animals? Given the negative feedback on AHR expression, reduced receptor levels might explain the lower Cyp1a1 expression in the CA-AhR model.

We have not assessed AHR protein levels in cells from the two groups. This would presumably only be possible via Western blot analysis which is experimentally difficult and not easily quantifiable in rare populations such as dendritic cells. Presumably the reviewer is referring to induction of Tiparp which might cause AHR degradation, however no induction was detected in the colon of CAIR mice or in TCDD treated mice 6 days after administration.

Theoretically the AHR-Tomato mutant could be crossed in with CAIR to look at this issue at the single cell level but apart from requiring a lengthy time of breeding, it is not likely that the degradation of the fluorescent label is comparable with that of AHR protein itself.

Overall, we would think that the lower levels of Cyp1a1 in CAIR are nevertheless orders of magnitude more pronounced in their persistence compared with normal transient AHR activation by physiological ligands.

-In EV1, fecal Lcn2 levels differ between WT and vehicle groups. This is unexpected and should be discussed. Could these differences influence the infection dynamics observed between TCDD- and CA-AhR-treated mice?

The previous Fig EV1 showed the response to DSS treatment. We have now removed all DSS data as we agreed with Reviewer 2 that the depth of analysis was insufficient for the conclusions we made.

Data Interpretation Issues

-The statement, "AhrdCAIR/dCAIR mice cleared the bacteria faster than wildtype mice (Fig.3A)" was not supported by the CFU/g faeces barplot from Fig. 3A, especially considering the unequal biological replicates between wild-type and CAIR mice.

We agree that this statement was exaggerated and have now modified the text.

-The authors stated that "AHR expression levels were higher in cells from the colon and the highest expression was seen in the myeloid lineage comprising dendritic cells, macrophages, and eosinophils". What statistical test was performed to prove that statement? Please provide SD bars in Figure 1.

The AHR-Tomato feature is a stable genetic modification and therefore not prone to variation in steady state like other measurements that require experimental manipulation. This is clearly indicated in the tight alignment of each of the two datapoints. It would have been difficult to increase the group size as the digestion protocols for isolation of some of the rarer cell types are sensitive and delays likely resulting in reduced cell viability.

We have modified Fig.1 to show not only MFI for Tomato but also percentage values.

Furthermore, we have included the gating strategy in Fig.S2. The histograms in Fig.S1 clearly show the higher expression levels in myeloid cells, and this has been described in an earlier study too (Diny et al.2022).

-In the "Methods" subsection "Statistics and data analysis", the authors state: "Statistically significant outliers were identified using the ROUT method (Q = 5%) and removed from the plots." This rule of thumb is alarming considering the potential abuse it permits, including the removal of biologically variation because of a perceived technical error.

We have included all datapoints that were previously removed to increase the transparency of our findings and have removed this statement from the Methods section.

-The statement that TCDD levels changed across tissues over 6 days needs statistical analysis (results; Fig. 2E). Please include appropriate tests to show that these differences are significant.

The levels for TCDD in serum and intestinal tissues were very low and we therefore were forced to pool the tissue samples of three mice for each datapoint. Hence, we cannot statistically prove that TCDD levels declined in these tissues over 6 days (although it is likely that TCDD transferred from gut to liver) and have therefore taken out the sentence stating this. We have rewritten

the results part to better stress this point. A supplementary table (Table S1) gives the individual values for TCDD in different tissues.

-The authors state that "Fig. 2F-H clearly shows that both models of prolonged AHR signaling exhibited clear toxic endpoints," including intrahepatic lipid accumulation. However, in Fig. 2H, there is no visible effect of TCDD or Ahr^{dCAIR/dCAIR} on lipid accumulation, as presented in the graph on the right. The authors should either include statistical analysis to support this claim or avoid drawing conclusions based solely on the visual appearance.

We apologise for this mistake and have corrected this in the text. While TCDD an initial experiment showed increased lipid accumulation, adding back an outlier and increasing the n made this trend not significant.

Experimental Design Clarity and Control Group Consistency

-In the "Mice" subsection of "Methods and Protocols", the strain of the wild-type mice is reported to be C57BL/6J and a reference to the Jackson laboratory is made in the "Reagents and Tools table". However, the background of the Ahr^{CAIR} strain is unclear from the manuscript alone, and the reference for the strain generation was also ambiguous. When directly comparing differences between TCDD or vehicle-treated wild-type and non-exposed CAIR mice (e.g., Fig. 2F - H), the "wild-type" mice should have the same genetic background as the mice with CAIR alleles. Otherwise, differences in strain background could explain differences between the CAIR and TCDD treated models as well as the non-exposed wild-type mice (e.g. Fig. 3A).

We have added a sentence in the Methods to emphasise that all mice used were of C57BL/6 background. As immunologists we are highly aware of the importance of genetic background in animal experiments.

-The authors state that they used either male or female mice, but their analyses do not distinguish results by sex, suggesting that they lumped together. Further clarification about the sex of the mice in this study is required.

We apologise for the lack of clarity with stating the sex of animals in experimental groups. This is now stated in all figure legends.

-The study design of the single-nuclear RNA-seq and ATAC-seq experiment needs clarification. For example, the first sentence of the subsection "Single-cell multiome sequencing of immune cells from the c-MLN and colon" in the "Methods" is garbled: "Four to six mice per group received an oral administration of with vehicle TCDD 6 days prior to infection with 2 x 10⁹ CFU C. rodentium as described above." This reader assumes there was a vehicle group to which the TCDD treated and CAIR mice were compared.

We apologise for the lack of clarity in this section. Additional details have been added in the respective Methods section. Ahr^{dCAIR/dCAIR} and vehicle mice received corn oil + DMSO as vehicle and both Ahr^{dCAIR/dCAIR} and TCDD were compared to a vehicle group.

-A visual summary of the experimental setup would enhance the clarity and reader understanding.

Single-Cell / Multiomic Analysis and Methods Concerns

-The authors performed unsupervised clustering in part determined using the FindMarkers method from Seurat. FindMarkers is not a reliable source for identifying DEGs because the p-values are inflated from treating cells as replicates and from "double-dipping" (i.e., using the same gene expression data that was used to cluster cells into cell types to calculate p-values for differentially expressed clustered- or cell-type specific genes). An alternative to DEG overrepresentation, such as gene set enrichment analysis (GSEA), would avoid the calculation of p-values. If the authors pursue an alternative method to calculate DEGs, they should consider reporting basic information on the DEGs (e.g., the number of DEGs per cell type and condition).

Unsupervised clustering based on the PCA from highly variable gene was used to identify cell populations in each dataset independently. A dataset for the comparison between treatment and control was created using the Seurat merge function, without further reprocessing. The FindMarkers function was then used to compare the same cell type between conditions, rather than between clusters which were themselves annotated using FindMarkers. This has been clarified in the methods section. The number of DEGs per cell type and condition are now included in Table S6.

We acknowledge a limitation of the project is unreplicated samples and that the methods used to identify affected genes will give higher false positive rates than a fully replicated design. Alternative methods such as DESeq2 or glmGamPoi still suffer from the design constraint, and would use cells as replicates, however.

We have additionally analysed the gene expression data using clusterProfiler's gseGO function to assess the association between genes and Gene Ontology biological pathway annotations. These results are included as a Fig. for the reviewer and are supportive of our enrichment analysis, with a higher proportion of genes involved in antigen processing and presentation pathways having a higher expression in dendritic cells from *Ahr^{dCAIR/dCAIR}* mice compared to respective cells from either Vehicle or TCDD treated mice. The results from GSEA were obtained by ranking genes based on the fold change of their aggregated expression, calculated using Seurat's FoldChange function. A limitation of this analysis is that due to the sparsity of the gene expression data, the number of genes with tied fold change is high, which affects the ranking and therefore the results. In addition, there is an unequal distribution of fold changes, which is visible in plots shown in the figure provided and affects the results of the analysis.

We have provided the data corresponding to antigen presentation and processing pathways for the reviewer, but we consider that the limitations mentioned above make this analysis less than ideal as the results can be misleading, and we decided to not include them in the final version of the manuscript. All corresponding GSEA tables can be provided if required.

Data Presentation and Figure Labeling Issues

-Please provide the TCDD levels reported in Fig 2E in a supplementary table. A brief comparison to previous studies using the same or similar designs in which TCDD was administered and measured in the same tissues should be added to corroborate the GC-MS method.

We appreciate this point raised by the reviewer. The TCDD levels have been added as a supplementary table.

While the GC-MS method used for analysis has not been previously published, it was developed and performed by an analytical chemist with strong experience in analysis of dioxins and other types of environmental pollutants in various tissues. The description of the methods used for the chemical tissue extraction and analysis for TCDD, I3C and ICZ that was previously included in the supplementary methods, has now been included in the methods in the main manuscript. Distribution to adipose and liver is well documented in rodent models (e.g. see the atsd report from 2024: <https://www.atsdr.cdc.gov/toxprofiles/tp104.pdf>). References demonstrating the levels in intestinal tissues over time using the same dose, route of exposure, and sampling time points we are however not aware of.

*-Figure 5C is a difficult read. The gene ontologies should be identified by their gene ontology code in the figure legend or in the manuscript text to enhance reproducibility.
-Figures 5 and EV5 contain panels labeled "CAIR," which is inconsistent with the rest of the manuscript where the model is referred to as AhrdCAIR/dCAIR. Additionally, "CAIR" is mentioned in the EV1 legend-please clarify whether this refers to AhrdCAIR/dCAIR or correct the labeling if it is an error.*

We apologize for the lack of clarity and have corrected this in both figure legends. The data corresponds to $Ahr^{dCAIR/dCAIR}$ mice. Full tables including the gene ontology codes have been included as supplementary tables (Table S2, S3).

-The distinction between panels A and B in EV2 is unclear. The figure legend implies they show the same data. Please revise the legend or clarify the intent of each panel.

We apologize for the lack of clarity. We have changed the order and composition of the supplementary figures and the new Fig.S3 shows bacterial clearance in female $Ahr^{dCAIR/dCAIR}$ or TCDD treated mice in A and B, whereas the previous figure EV2B, now Fig.S4, shows the delayed clearance of bacteria beyond day 14 in TCDD treated male mice (corresponding to Fig.3).

Methodological Details / Reagents and Protocols

-For the subsection "NP-Ficoll and NP-CGG immunizations and measurement of anti-NP antibodies" in "Methods", a brief explanation for why multiple serum dilutions were used is requested to provide context, especially for readers without a background in immunology.

We appreciate this point and have now included an explanatory sentence in the Methods.

-Cell isolation: What is the rationale of adding FBS to digestion buffer? It is well known that FBS inhibits enzymatic activity of collagenase, and it is typically added when resuspending the cells after digestion to stop collagenase activity.

The reviewer is correct stating that FBS will inhibit the enzymatic activity of collagenase, but the addition of FBS drastically improved cell viability, especially for myeloid cells, which were the focus of the analysis in the Multiome.

Other

-The supplementary appendix was not included and thus could not be evaluated. Please ensure all supplementary files are provided for peer review.

We are surprised that the reviewer did not have access to these files as Reviewer 2 referred to data in the supplementary files so that we assume they must have been available.

GSEA - Antigen processing and presentation in dendritic cells

A Colon - cDC2 - *Ahr*^{dCAIR/dCAIR} vs Vehicle

Antigen proc and pres of exogenous peptide antigen
GO:0002478

Antigen proc and pres of endogenous antigen
GO:0019883

B Colon - cDC2 - *Ahr*^{dCAIR/dCAIR} vs TCDD

Antigen proc and pres of exogenous antigen
GO:0019884

Antigen proc and pres of endogenous antigen
GO:0019883

Colon - Activated cDC - *Ahr*^{dCAIR/dCAIR} vs TCDD

Antigen processing and presentation
GO:0019882

Antigen proc and pres of exogenous peptide antigen
GO:0002478

Ag proc and pres of exogenous antigen
GO:0019884

c-MLN - Activated cDC2 - *Ahr*^{dCAIR/dCAIR} vs TCDD

Antigen processing and presentation
GO:0019882

Antigen proc and pres of exogenous antigen
GO:0019884

Antigen proc and pres of endogenous peptide antigen
GO:0002483

Gene set enrichment analysis (GSEA) in dendritic cells.

GSEA for pathways related to antigen processing in (A) *Ahr*^{dCAIR/dCAIR} mice vs Vehicle and (B) *Ahr*^{dCAIR/dCAIR} mice vs TCDD. Genes were ranked by log₂ FC and pathways shown have an adjusted p-value <0.01. Red line marks maximum ES. Abbreviations: proc: processing; pres: presentation; ES: enrichment score.

Method description: For GSEA analyses, the log2 fold change of comparison datasets was calculated and normalised, within cell types, using the FoldChange function of Seurat. The log2 fold change between experimental and control datasets was used to rank the unfiltered gene sets. The gseGO function from clusterProfiler was used to test for enrichment using the "biological processes" subontology, a minimum gene set size of 30, p-value cutoff of 0.01 and multiple testing correction using the Benjamini-Hochberg method. The lists of enriched pathways involved in antigen presentation showed in the previous figure is presented below.

Enrichment in Ahr^{dCAIR/dCAIR} with respect to Vehicle group

ID	Description	setSize	enrichmentScore	NES	pvalue	p.adjust	qvalue	rank	leading_edge	Cell type
GO:0002478	antigen processing and presentation of exogenous peptide antigen	50	-0.449914689	-2.873444	1.19E-10	2.35E-08	2.06E-08	17783	tags=100%, list=55%, signal=45%	Colon_cDC2
GO:0019883	antigen processing and presentation of endogenous antigen	42	-0.365431788	-2.228956	3.08E-06	9.5E-05	8.31E-05	17119	tags=93%, list=53%, signal=44%	Colon_cDC2
GO:0002478	antigen processing and presentation of exogenous peptide antigen	50	-0.393078112	-2.243653	4.11E-05	0.000627	0.000429	17625	tags=96%, list=55%, signal=44%	c-MLN_Activated cDC2
GO:0048002	antigen processing and presentation of peptide antigen	75	-0.325404903	-2.025432	8.72E-05	0.00115	0.000786	17037	tags=89%, list=53%, signal=42%	c-MLN_Activated cDC2
GO:0019883	antigen processing and presentation of endogenous antigen	42	-0.351543727	-1.916806	0.00083	0.006685	0.004571	17037	tags=90%, list=53%, signal=43%	c-MLN_Activated cDC2

Enrichment in Ahr^{dCAIR/dCAIR} with respect to TCDD group

ID	Description	setSize	enrichmentScore	NES	pvalue	p.adjust	qvalue	rank	leading_edge	Cell type
GO:0019884	antigen processing and presentation of exogenous antigen	56	-0.447330044	-3.080332	1.34E-10	2.75E-08	2.32E-08	17869	tags=100%, list=55%, signal=45%	Colon_cDC2
GO:0019883	antigen processing and presentation of endogenous antigen	42	-0.322985	-1.891059	0.000143	0.002128	0.001799	17201	tags=90%, list=53%, signal=42%	Colon_cDC2
GO:0019882	antigen processing and presentation	124	0.38797768	2.033915	2.7E-05	0.000163	6.04E-05	5131	tags=39%, list=16%, signal=33%	Colon_Activated cDC
GO:0002478	antigen processing and presentation of exogenous peptide antigen	50	0.463636976	2.05627	0.00062	0.002181	0.000809	5443	tags=48%, list=17%, signal=40%	Colon_Activated cDC
GO:0019884	antigen processing and presentation of exogenous antigen	56	-0.324991569	-1.958992	0.000459	0.007386	0.005659	17988	tags=93%, list=56%, signal=41%	c-MLN_Activated cDC1
GO:0019882	antigen processing and presentation	124	-0.234435477	-1.804174	3.62E-05	0.000719	0.000568	18004	tags=89%, list=56%, signal=39%	c-MLN_Activated cDC2
GO:0019884	antigen processing and presentation of exogenous antigen	56	-0.370451963	-2.331836	1.78E-05	0.00042	0.000332	18580	tags=96%, list=58%, signal=41%	c-MLN_Activated cDC2
GO:0002483	antigen processing and presentation of endogenous peptide antigen	40	-0.34697085	-1.969411	0.000449	0.005175	0.004087	18004	tags=92%, list=56%, signal=41%	c-MLN_Activated cDC2

September 9, 2025

RE: Life Science Alliance Manuscript #LSA-2025-03414-TR

Dr. Brigitta Stockinger
The Francis Crick Institute
1 Midland Road
London, London NW1 1AT
United Kingdom

Dear Dr. Stockinger,

Thank you for submitting your revised manuscript entitled "Beneficial and detrimental consequences of AHR activation in intestinal infection".

Your revised manuscript was evaluated by two of the three original reviewers whose comments are appended below. The two reviewers note that their previous concerns are addressed in this revised version. We concur with the reviewers and would be happy to publish your paper in Life Science Alliance pending final revisions necessary to meet our formatting guidelines.

- Please clarify the statement in the DA section, "Supplementary AHR KO snMultiome datasets from both colon and c-MLN are made available (data not shown)". Do you mean that this dataset is made available as supplemental information or will be made available upon request? Please remove 'data not shown' from the statement.
- Please provide citation for the mice described in the methods section.
- Please refer to the reagents and tools tables in the methods section of the manuscript/table.
- Please upload your Tables in editable .doc or Excel format.
- Please add the X and Bluesky handles of your host institute/organisation, as well as your own and/or one of the authors, in our system.
- There is a name discrepancy for one of your co-authors, please correct: Oscar Perez Diaz in the manuscript file vs. Oscar E. Diaz in the system.
- Please mark Dr. Emma Wincent as a secondary corresponding author in our system as well.
- Please add your main, supplementary figure, and table legends to the main manuscript text after the references section.
- Please add an Author Contributions section to your main manuscript text.
- Please remove legends from the supplementary figures. They should appear only in the manuscript file.
- Please add callouts for Figures 1A-B; S1A-B and S2A-B to your main manuscript text.
- Please be sure that the authorship listing and order is correct.

A. FINAL FILES:

B. MANUSCRIPT ORGANIZATION AND FORMATTING:

Thank you for your attention to these final processing requirements. Please revise and format the manuscript and upload materials as soon as you are able.

Sincerely,

Sarita Hebbar, PhD
Scientific Editor
Life Science Alliance
<http://www.lsjournal.org>

Reviewer #1 (Comments to the Authors (Required)):

I appreciate the thoughtful responses of the authors. This reviewer has no further requests for changes and will be using data from the ms for years to come.

Reviewer #3 (Comments to the Authors (Required)):

The authors have adequately addressed all concerns raised in the review.

September 12, 2025

RE: Life Science Alliance Manuscript #LSA-2025-03414-TRR

Dr. Brigitta Stockinger
The Francis Crick Institute
1 Midland Road
London, London NW1 1AT
United Kingdom

Dear Dr. Stockinger,

Thank you for submitting your Research Article entitled "Beneficial and detrimental consequences of AHR activation in intestinal infection". It is a pleasure to let you know that your manuscript is now accepted for publication in Life Science Alliance. Congratulations on this interesting work.

DISTRIBUTION OF MATERIALS:

Again, congratulations on a very nice paper. I hope you found the review process to be constructive and are pleased with how the manuscript was handled editorially. We look forward to future exciting submissions from your lab.

Sincerely,

Sarita Hebbar, PhD
Scientific Editor
Life Science Alliance
<http://www.lsajournal.org>